# ADVERSARIAL COLLABORATIVE LEARNING ON NON-IID FEATURES

## ABSTRACT

Federated learning has been a popular approach to enable collaborative learning on multiple parties without exchanging raw data. However, the model performance of federated learning may degrade a lot due to non-IID data. While most existing studies focus on non-IID labels, federated learning on non-IID features has largely been overlooked. Different from typical federated learning approaches, the paper proposes a new learning concept called ADCOL (**Ad**versarial **Co**llaborative **L**earning) for non-IID features. Instead of adopting the widely used model-averaging scheme, ADCOL conducts training in an adversarial way: the server aims to train a discriminator to distinguish the representations of the parties, while the parties aim to generate a common representation distribution. Our experiments on three real-world datasets show that ADCOL achieves better accuracy and is much more communication-efficient than state-of-the-art federated learning algorithms on non-IID features. More importantly, ADCOL points out a promising research direction for collaborative learning.

## 1 INTRODUCTION

Deep learning is data hungry. While data are always dispersed in multiple parties (e.g., mobile devices, hospitals) in reality, data are not allowed to transfer to a central server for training due to privacy concerns and data regulations. Collaborative learning among multiple parties without the exchange of raw data has been an important machine learning topic.

Federated learning (McMahan et al., 2016; Kairouz et al., 2019; Li et al., 2019b;a) has been a popular form of collaborative learning without exchanging raw data. A basic federated learning framework is FedAvg (McMahan et al., 2016), which uses a model-averaging scheme. In each round, the parties update their local models and send them to the server. The server averages all local models to update the global model, which is sent back to the parties as the new local model in the next round. FedAvg has been widely used due to its effectiveness and simpleness. Most existing federated learning approaches are designed based on FedAvg.

However, as shown in many existing studies (Hsu et al., 2019; Li et al., 2020; Wang et al., 2020b; Karimireddy et al., 2020; Acar et al., 2021; Li et al., 2021b), the performance of FedAvg and its alike algorithms may be significantly degraded in non-IID data among parties. While many studies try to improve FedAvg on non-IID data, most of them (Li et al., 2020; Wang et al., 2020b; Karimireddy et al., 2020; Acar et al., 2021; Li et al., 2021a; Wang et al., 2020a) focus on the label imbalance setting, where the parties have different label distributions. In their experiments, they usually simulate the federated setting by unbalanced partitioning the dataset into multiple subsets according to the labels.

As summarized in (Hsieh et al., 2020; Kairouz et al., 2019), besides the label distribution skew, feature imbalance is also an important case of non-IID data. In the feature imbalance setting, the feature distribution $P_i(\mathbf{x})$ varies across parties. This setting widely exists in reality, e.g., people have different stroke width and slant when writing the same word. Another example in practice is that medical images of different hospitals have different intensity and contrast. However, compared with non-IID labels, federated learning on non-IID features has been less explored. Since the feature distributions are different across parties, there exists bias in the local training. It is challenging to mitigate the bias while exploiting the heterogeneous feature distributions in the training.

In this paper, we think out of the model-averaging scheme used in federated learning, and propose a novel learning concept called adversarial collaborative learning. While the feature distribution of each party is different, to mitigate the local bias, we aim to learn a common representation distribution among parties. Specifically, the server aims to train a discriminator to distinguish the local representations by the party IDs, while the parties train the local models such that the generated representations cannot be distinguished by the discriminator. Our experiments show that ADCOL outperforms state-of-the-art federated learning algorithms (Li et al., 2021b; 2020; Fallah et al., 2020; Collins et al., 2021) on three real-world tasks. More importantly, ADCOL points out a promising research direction on collaborative learning. Inspired by our idea of conducting collaborative learning in an adversarial way, we believe there will be more interesting future works in this direction.

## 2 BACKGROUND AND RELATED WORK

### 2.1 NON-IID DATA

We use $P_i(\mathbf{x}, y)$ to denote the data distribution of party $i$, where $\mathbf{x}$ is the features and $y$ is the label. According to existing studies (Kairouz et al., 2019; Hsieh et al., 2020), we can categorize non-IID data in federated learning into the following four classes: (1) non-IID labels: the marginal distribution $P_i(y)$ varies across parties. (2) non-IID features: the marginal distribution $P_i(\mathbf{x})$ varies across parties. (3) concept drift: The conditional distributions $P_i(y|\mathbf{x})$ or $P_i(\mathbf{x}|y)$ varies across parties. (4) quantity skew: the amount of data varies across parties. In this paper, we focus on non-IID features, which widely exist in reality. For example, while the pneumonia distribution of patients may be close among different hospitals, the distributions of CT images (such as resolution, color, length and width) may vary across hospitals, which are collected from different equipment and environments.

### 2.2 FEDERATED LEARNING ON NON-IID LABELS

Non-IID data is a key challenge in federated learning. There have been many studies trying to improve the performance of federated learning under non-IID data. However, most existing studies (Li et al., 2020; Wang et al., 2020a; Hsu et al., 2019; Li et al., 2021a; Acar et al., 2021; Karimireddy et al., 2020) focus on label imbalance, where they usually simulate the federated setting with heterogeneous label distributions in the experiments. For example, FedProx (Li et al., 2020) introduces a proximal term in the objective of local training, which limits the update of the local model by the distance between the local model and the global model. FedMA (Wang et al., 2020a) applies Bayesian non-parametric methods to update the global model in a layer-wise manner in the model aggregation phase. FedDyn (Acar et al., 2021) introduces a dynamic regularization term in the local training based on the global model and the local model from the previous round. More recently, MOON (Li et al., 2021a) proposes model-contrastive federated learning, which corrects the local training by a contrastive loss which maximizes the agreement between the representation learned by the global model and the representation learned by the local model. As we will show in the experiments, existing algorithms have severe performance degradation on parties with non-IID features. More recently, a contemporary study (Zhang et al., 2021a) proposed FedUFO, where each party trains a discriminator to apply feature and objective consistency constrains to address the non-IID data issue. However, FedUFO needs to transfer each local model to all the other parties, which causes massive communication overhead. Moreover, FedUFO is still based on FedAvg, while we propose a novel new collaborative training scheme based on adversarial training in this paper.

### 2.3 FEDERATED LEARNING ON NON-IID FEATURES

To the best of our knowledge, there is only one study named FedBN (Li et al., 2021b) that focuses on non-IID features. FedBN introduces lightweight modifications on FedAvg. Instead of transferring and averaging all model parameters in FedAvg, FedBN keeps all the local batch normalization parameters and does not synchronize them with the global model. The operations for non-batch normalization parameters are the same as FedAvg. Our proposed ADCOL is based on a novel approach: adversarial learning for a common representation among multiple parties. ADCOL outperforms FedBN on three real-world datasets in both accuracy and communication-efficiency.

## 2.4 PERSONALIZED FEDERATED LEARNING

There are basically two kinds of objective in federated learning: learning a single global model or learning a personalized local model for each party (i.e., personalized federated learning (Fallah et al., 2020; Dinh et al., 2020; Hanzely et al., 2020; Zhang et al., 2021b; Huang et al., 2021; Collins et al., 2021)). When the training data is non-IID across parties, personalized federated learning is reasonable since a single global model may not be good for every local distribution. In this paper, same as FedBN, ADCOL is designed under the personalized federated learning setting, i.e., each party aims to learn a local model for its own local task.

## 3 METHOD

### 3.1 PROBLEM STATEMENT

Suppose there are $N$ parties, where party $i$ has a local dataset $\mathcal{D}^i = \{\mathbf{x}^i, y^i\}$. The feature distributions $P(\mathbf{x}^i)$ are different among parties while the label distributions $P(y^i)$ are similar among parties. The parties conduct collaborative learning over $\mathcal{D} \triangleq \bigcup_{i \in [N]} \mathcal{D}^i$ with the help of a central server without exchanging the raw data. The goal of each party is to train a machine learning model which has good accuracy on its local test dataset.

### 3.2 MOTIVATION

The key idea of ADCOL is intuitive. While the feature distributions are different among parties, to mitigate the local bias, ADCOL aims to learn a common task-specific representation distribution. Inspired by GANs (Goodfellow et al., 2014) and adversarial learning for distribution matching (Tzeng et al., 2017; Peng et al., 2019b), ADCOL adopts an adversarial way to restrict the representation distribution.

Next we explain the motivation from a distribution perspective. The ultimate goal of each party in typical federated learning is to learn the oracle optimal distribution $p^*(y|\mathbf{x})$. Here we introduce the representation $\mathbf{z}$ of the data $\mathbf{x}$, which is supposed to be sufficient for the prediction of labels $y$ (Federici et al., 2020). Then, $\mathbf{x}$, $y$, and $\mathbf{z}$ forms a Markov chain $\mathbf{x} \to \mathbf{z} \to y$. Thus, we decomposite the oracle optimal into two parts: $p^*(\mathbf{z}|\mathbf{x})$ and $p^*(y|\mathbf{z})$. The first part is to learn a good representation and the second part is to predict the label by the representation with simple linear layers. To learn the optimal representation distribution $p^*(\mathbf{z}|\mathbf{x})$, the objective of each party is

$$\min_{\theta_i} E_{\mathbf{x} \sim D_i} \ell_{KL}(p(\mathbf{z}|\mathbf{x}; \theta_i) \,||\, p^*(\mathbf{z}|\mathbf{x})), \tag{1}$$

where $\ell_{KL}$ is the KL divergence loss. We adopt KL divergence due to its popularity for evaluating the distance between probability distributions. However, the optimal representation distribution is unknown. Thus, we approximate the objective by two aspects: 1) By training in a novel adversarial way, we use the similarity between the representations of different parties to regularize the representation, i.e., the parties learn a common representation $\mathbf{z}$. 2) We ensure that the generated representation contains necessary information for the prediction of $y$ by training on the labeled local data. Next, we describe our model architecture and training algorithm in details.

### 3.3 MODEL ARCHITECTURE

There are two kinds of models in ADCOL: the local models trained in the parties and the discriminator trained in the server. As ADCOL works from the perspective of representation, the architecture of the local model is similar as existing studies (Chen et al., 2020; He et al., 2020; Chen and He, 2021) on self-supervised representation learning. The local model has three components: a base encoder, a projection head, and a predictor. The base encoder (e.g., ResNet-50) extracts representation vectors from inputs. Like SimCLR (Chen et al., 2020) and SimSam (Chen and He, 2021), an additional projection head is introduced to map the representation to a space with a fixed dimension. The final predictor is used to output predicted probabilities for each class. For ease of presentation, we use $F(\cdot)$ to denote the whole model and $G(\cdot)$ to denote the model before the predictor (i.e., $G(\mathbf{x})$ is the mapped representation vector of input $\mathbf{x}$). For the discriminator, we simply use a multi-layer perceptron in our experiments.

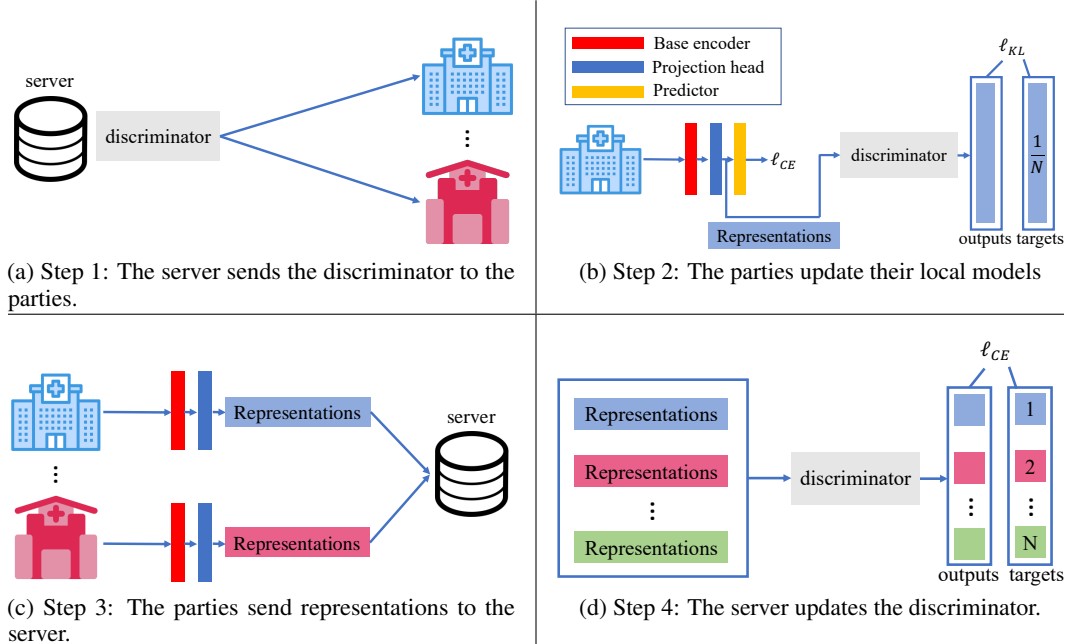

(a) Step 1: The server sends the discriminator to the parties.

(b) Step 2: The parties update their local models

(c) Step 3: The parties send representations to the server.

(d) Step 4: The server updates the discriminator.

Figure 1: The ADCOL framework

## 3.4 The Overall Framework

The overall framework is shown in Figure 1 and Algorithm 1. There are four steps in each round: (1) The server sends the discriminator to the parties. (2) The parties update their local models. (3) The parties send representations to the server. (4) The server updates the discriminator.

**Step 1** In the first step, the server sends the discriminator to parties (line 4 of Algorithm 1).

**Step 2** In the second step, the parties update their models using their local datasets (lines 10-17 of Algorithm 1). In addition to the objective which aims to minimize the cross-entropy loss (i.e., $\ell_{CE}$) on the local dataset, ADCOL introduces an additional regularization term which aims to maximize the probability that the discriminator cannot distinguish the local representations. For each input $\mathbf{x}$, ADCOL feeds the representation $G(\mathbf{x})$ to the discriminator. ADCOL expects the discriminator to output probability vector $[\frac{1}{N}]^N$ (i.e., the probability of each class is $\frac{1}{N}$) such that it cannot distinguish which party that the representation comes from. Thus, ADCOL uses Kullback–Leibler (KL) divergence loss to measure the difference between the output of the discriminator $D(G(\mathbf{x}))$ and the target $[\frac{1}{N}]^N$. The final loss of an input $(\mathbf{x}, y)$ is computed as

$$\ell = \ell_{CE}(F(\mathbf{x}), y) + \mu\ell_{KL}([\frac{1}{N}]^N \parallel D(G(\mathbf{x}))) \tag{2}$$

where $\mu$ is a hyper-parameter to control the weight of KL divergence loss, $\ell_{CE}$ is the cross-entropy loss, and $\ell_{KL}$ is the KL divergence loss.

**Step 3** After local training, the parties feed their data into the local models and transfer the representations to the server (line 5 of Algorithm 1).

**Step 4** The server updates the discriminator using the received representations (lines 6-9 of Algorithm 1). Specifically, the server can build a training set, where the feature values are the representations and the labels are the party IDs that the representations come from. The training set is used to update the discriminator using stochastic gradient descent.

---

**Algorithm 1:** The ADCOL algorithm

---

**Input:** number of communication rounds $T$, number of parties $N$, number of local epochs $E$,
learning rate $\eta$, hyper-parameter $\mu$
**Output:** The local models $F_i$ ($i \in [N]$)

---

1 **Server executes**:
2 **for** $t = 1, 2, ..., T$ **do**
3     **for** $i = 1, 2, ..., N$ **in parallel do**
4        send the discriminator $D$ to party $i$
5        $\mathbf{R}_i \leftarrow$ **PartyLocalTraining**$(i, D)$
6     $\mathbf{R} \leftarrow \{(\mathbf{R}_i, i)\}_{i=1}^{N}$
7     **for** each batch $\mathbf{b} = \{\mathbf{R}_i, i\}$ of $\mathbf{R}$ **do**
8        $\ell \leftarrow CrossEntropyLoss(D(\mathbf{R}_i), i)$
9        $D \leftarrow D - \eta\nabla\ell$

10 **PartyLocalTraining**$(i, D)$:
11 **for** epoch $e = 1, 2, ..., E$ **do**
12     **for** each batch $\mathbf{b} = \{\mathbf{x}, y\}$ of $\mathcal{D}^i$ **do**
13        $\ell_{CE} \leftarrow CrossEntropyLoss(F_i(\mathbf{x}), y)$
14        $\mathbf{R} \leftarrow G_i(\mathbf{x})$
15        $\ell_{KL} \leftarrow KLDiv([\frac{1}{N}]^N || D(\mathbf{R}))$
16        $\ell \leftarrow \ell_{CE} + \mu\ell_{KL}$
17        $F_i \leftarrow F_i - \eta\nabla\ell$
18 return $G_i(\mathbf{x}^i)$ to server

---

### 3.5 COMPARISONS WITH EXISTING FEDERATED LEARNING APPROACHES

ADCOL is quite different from typical federated learning algorithms. Instead of transferring and averaging models, ADCOL transfers the representations from the parties to the server. More importantly, the training is conducted in an adversarial way between the parties and the server. The server tries to distinguish the representations while the parties try to generate non-distinguishable representations. Here we compare ADCOL with FedAvg in terms of communication size and privacy.

**Communication Size** For simplicity, we do not consider party sampling during training (i.e., all parties participate in learning in each round). We use $S_L$ to denote the size of the local model. Then, the communication size of FedAvg is $2NS_L$, including the server sends the model to all parties and the parties send their local models to the server. We use $n$ to denote the total number of examples (i.e., $n = \sum_{i=1}^{N} |\mathcal{D}^i|$), $d$ to denote the dimension of the representations, and $S_D$ to denote the size of the discriminator. Suppose each float value costs 4 bytes to store. In each round, the total communication size of ADCOL is $(4nd + NS_D)$, including the parties send the representations to the server and the server sends the discriminator to the parties. Thus, when $n > (2S_L - S_D)N/4d$, the communication size of ADCOL is larger than FedAvg in each round.

Note that the above threshold is usually large in practice. For example, in our experiments, when $N = 6$, $d = 2048$, $S_L = 142MB$, and $S_D = 5MB$, the threshold is equal to 204346. As we will show in the experiments, ADCOL is much more communication-efficient than FedAvg. Moreover, ADCOL requires a smaller bandwidth requirement as the size of each representation is much smaller than the local model.

**Privacy** While FedAvg transfers models and ADCOL transfers the representations, FedAvg and ADCOL both suffer from possible attacks on the exchanged information. There are many existing studies (Shokri et al., 2017; Nasr et al., 2019) that infer sensitive information from exchanged gradients/models. Also, there are studies (Salem et al., 2020) on reconstruction attacks wn the output of a model. While existing studies have shown that the mutual information between the input data and the final representation is small (Shwartz-Ziv and Tishby, 2017), it is not clear that which one is more private against inference attacks, which can be an interesting future direction. Local differential privacy (Kairouz et al., 2014) can be applied to protect the representation, which has been adopted in previous studies (Lyu et al., 2020). We have conducted experiments to show the feasibility of differential privacy to protect the communicated representations in Appendix B.13.

## 4 THEORETICAL ANALYSIS

As shown in Equation (2), the local loss has two parts: the cross-entropy loss part to update the whole network $F$ and the KL divergence loss part to update the representation generator $G$. Ideally, to achieve minimum of $\ell$, each part should achieve minimum. Since the cross-entropy loss part is same as FedAvg, we focus on the effect of the KL divergence loss. For simplicity, we ignore the

cross-entropy loss and study the KL divergence loss in our theoretical analysis[1]. The local objective of party $i$ is:

$$\min_{G_i} \mathbb{E}_{\mathbf{x} \sim \mathcal{D}_i} \ell_{KL}([\frac{1}{N}]^N \parallel D(G_i(\mathbf{x}))) \tag{3}$$

The objective of the discriminator is

$$\max_D \sum_{i=1}^{N} \mathbb{E}_{\mathbf{x} \sim \mathcal{D}_i} \log(D_i(G_i(\mathbf{x}))), \tag{4}$$

where $D_i(\cdot)$ is the $i$-th output of the prediction vector $D(\cdot)$ (i.e., the probability of class $i$). Here we analyze the convergence property of the training process. All the proofs are available in Appendix A.

**Theorem 1.** We use $P_{G_i}$ to denote the distribution of the representations generated in party $i$ and $P_{G_i}(\mathbf{z})$ is the probability of representation $\mathbf{z}$ in distribution $P_{G_i}$. Then, the optimal discriminator $D^*$ of Equation (4) is

$$D_k^*(\mathbf{z}) = \frac{P_{G_k}(\mathbf{z})}{\sum_{i=1}^{N} P_{G_i}(\mathbf{z})}. \tag{5}$$

**Theorem 2.** Given the optimal discriminator $D^*$ from Equation (5), the global minimum of Equation (3) is achieved if and only if

$$P_{G_1} = P_{G_2} = \cdots = P_{G_N} \tag{6}$$

Theorem 1 and Theorem 2 show that to achieve the minimum of the objectives of the local parties and the discriminator, the parties will generate the same representation distribution, which matches the goal of ADCOL.

**Theorem 3.** Suppose $P_G^*$ is the optimal solution shown in Theorem 2. If $G_i$ ($\forall i \in [1, N]$) and $D$ have enough capacity, and $P_{G_i}$ is updated to minimize the local objective (i.e., Equation (3)), given the optimal discriminator $D^*$ from Equation (5), then $P_{G_i}$ converges to $P_G^*$.

The above theorem provides insights on the convergence of the training. In practice, we optimize the parameter $\theta$ of the local networks rather than $P_{G_i}$ itself, which is reasonable due to the excellent performance as claimed in (Goodfellow et al., 2014).

Note that there are collapsing solutions for Equation (3) and Equation (4). The representations of each party can simply be constant vectors, which can achieve global minimum of Equation (3). Thus, the cross-entropy loss is necessary in Equation (2), which ensures that the generated representations are meaningful.

## 5 EXPERIMENTS

### 5.1 EXPERIMENTAL SETUP

**Baselines** We compare ADCOL with six baselines including SOLO (i.e., each party trains the model individually without collaborative learning), FedAvg (McMahan et al., 2016), FedBN (Li et al., 2021b), FedProx (Li et al., 2020), Per-FedAvg (Fallah et al., 2020), and FedRep (Collins et al., 2021). Here FedBN is the state-of-the-art FL approach on non-IID features. FedProx is a popular FL approach for non-IID data. Per-FedAvg and FedRep are two state-of-the-art personalized FL approaches. Like FedAvg (McMahan et al., 2016), we use weighted average according to the data volume of each party for all baselines.

**Models** All approaches use the same local model architecture for a fair comparison. The architecture of the local model is similar as SimSam (Chen and He, 2021), which has the following three components: (1) Base encoder: ResNet-50 (He et al., 2016). (2) Projection head: a 3-layer MLP with BN applied to each fully-connected layer. The input dimension is 4096. The dimension of the hidden

---

[1]Note that $G$ is a part of $F$ and the two losses are not independent of each other. For simplicity, we only analyze the KL divergence loss to study its effect.

Table 1: The comparison of top-1 test accuracy among different approaches on Digits. We run FL approaches for 100 rounds (all approaches have converged). We run three trials and report the mean and standard derivation. Besides the test accuracy on each party, we also report the mean accuracy of all parties denoted as "AVG".

| Digits | MNIST | SVHN | USPS | SynthDigit | MNIST_M | AVG |
|---|---|---|---|---|---|---|
| SOLO | 87.9%±0.4% | 34.8%±0.8% | 94.8%±0.1% | 63.0%±0.4% | 67.2%±0.4% | 69.5%±0.3% |
| FedAvg | 94.4%±0.5% | 59.4%±0.9% | 94.3%±0.2% | 74.4%±0.5% | 70.3%±1.2% | 78.6%±0.6% |
| FedBN | 94.1%±0.8% | **59.9%**±0.7% | 94.1%±0.1% | 73.9%±0.6% | 71.3%±1.1% | 78.7%±0.6% |
| FedProx | 94.1%±0.4% | 59.8%±0.6% | 94.3%±0.1% | 73.4%±0.3% | 71.6%±0.9% | 78.6%±0.4% |
| Per-FedAvg | 88.9%±0.7% | 36.6%±1.3% | 89.5%±0.2% | 58.3%±0.7% | 54.5%±1.3% | 65.6%±0.8% |
| FedRep | 92.6%±0.2% | 42.0%±1.0% | 93.1%±0.1% | 61.1%±0.5% | 50.8%±1.4% | 67.9%±0.8% |
| ADCOL | **94.7%**±0.6% | 58.2%±1.0% | **95.4%**±0.2% | **76.0%**±0.3% | **76.7%**±0.8% | **80.2%**±0.5% |

layer and the output layer is 2048. (3) Predictor: a 2-layer MLP with BN applied to its hidden layer. The input dimension is 2048. The dimension of its hidden layer is 512. The discriminator is a 3-layer MLP. The input dimension is 2048. The dimension of the hidden layers is 512. The output dimension is equal to the number of parties.

**Datasets** We use the same datasets as in the study of FedBN. There are three real-world tasks in our experiments[2]: (1) *Digits*: The Digits task has the following five digit data sources from different domains: MNIST (LeCun et al., 1998), SVHN (Netzer et al., 2011), USPS (Hull, 1994), SynthDigits (Ganin and Lempitsky, 2015), and MNIST-M (Ganin and Lempitsky, 2015). (2) *Office-Caltech-10* (Gong et al., 2012): The dataset has four data sources acquired using different camera devices or in different real environments with various backgrounds: Amazon, Caltech, DSLR, and WebCam. (3) *DomainNet* (Peng et al., 2019a): The dataset contains natural images coming from six different data sources with different image styles: Clipart, Infograph, Painting, Quickdraw, Real, and Sketch.

**Setup** By default, the number of parties is equal to the number of data sources, where each party has data from one of the data sources. For each dataset, we randomly split 1/5 of the original dataset as the test dataset, while the remained dataset is used as the training dataset. Same as (Li et al., 2021b), data of different parties have heterogeneous features but share the same labels and label distribution, which naturally forms the non-IID feature setting. The number of local epochs is set to 10 by default for all FL approaches. The number of epochs is set to 300 for SOLO. For ADCOL and FedProx, we tune $\mu \in \{10, 1, 0.1, 0.01, 0.001\}$ and report the best results. For FedRep, we tune $\beta$ (i.e., step size for the second batch training) from 0.001, 0.01 and report the best results. We use the prediction layers as the shared representation in FedRep. We use PyTorch v1.81 (Paszke et al., 2019) to implement all approaches. We use the SGD optimizer for training with a learning rate of 0.01. The SGD weight decay is set to $10^{-5}$ and the SGD momentum is set to 0.9. The batch size is set to 64, 32, and 32 for Digits, Office-Caltech-10, and DomainNet, respectively. We run the experiments on a server with 8 * NVIDIA GeForce RTX 3090 and a cluster with 45 * NVIDIA GeForce RTX 2080 Ti.

Due to the page limit, we only present some representative experimental results in the main paper. For more experimental results and details, please refer to Appendix B.

## 5.2 OVERALL COMPARISON

Table 1 reports the test accuracy of different approaches on three tasks. We have the following observations. First, ADCOL is more effective than the other approaches. It can achieve the best test accuracy on most datasets. Moreover, ADCOL can outperform the other approaches by more than 2% accuracy on average. Second, while the parties may not benefit from FL approaches in some cases (e.g., Caltech-10), ADCOL always achieves better accuracy than SOLO, which demonstrates the robustness of ADCOL. Last, the personalized FL approaches (i.e., Per-FedAvg and FedRep) have a poor performance on the non-IID feature setting, which are even worse than SOLO.

---

[2]FedBN uses four tasks in their experiments. We only use three tasks because one task is not publicly available in FedBN's GitHub website.

Table 2: The communication round and communication cost of each approach to achieve the same target performance as the minimum converged accuracy among FedAvg, FedBN, FedProx, and ADCOL as shown in Table 1 (i.e., 94.1% in MNIST). We use the slash cell to indicate that the approach cannot reach the target performance in 100 rounds/30 GB. The speedup is computed by dividing the communication size of FedAvg by the communication size of ADCOL.

| Digits | | MNIST | SVHN | USPS | SynthDigit | MNIST_M | AVG |
|---|---|---|---|---|---|---|---|
| #round | FedAvg | 11 | 54 | 5 | 11 | 7 | 28 |
| | FedBN | 11 | 73 | 5 | 68 | 7 | 22 |
| | FedProx | 64 | 42 | 8 | 12 | 10 | 31 |
| | Per-FedAvg | ╲ | ╲ | ╲ | ╲ | ╲ | ╲ |
| | FedRep | ╲ | ╲ | ╲ | ╲ | ╲ | ╲ |
| | ADCOL | 19 | 86 | 6 | 19 | 9 | 21 |
| cost (GB) | FedAvg | 3.12 | 15.34 | 1.42 | 3.12 | 1.99 | 7.95 |
| | FedBN | 3.12 | 20.73 | 1.42 | 19.31 | 1.99 | 6.25 |
| | FedProx | 18.18 | 11.93 | 2.27 | 3.41 | 2.84 | 8.80 |
| | Per-FedAvg | ╲ | ╲ | ╲ | ╲ | ╲ | ╲ |
| | FedRep | ╲ | ╲ | ╲ | ╲ | ╲ | ╲ |
| | ADCOL | 0.21 | 0.95 | 0.07 | 0.21 | 0.10 | 0.23 |
| Speedup | | **14.95** | **16.21** | **21.52** | **14.95** | **20.08** | **34.42** |

## 5.3 COMMUNICATION EFFICIENCY

To show the communication efficiency of ADCOL, like existing studies (Karimireddy et al., 2020; Lin et al., 2020), we compare the number of communication rounds and communication size of each approach to achieve the same target performance. The results on Digits are shown in Table 2. The results on the other two datasets are available in Appendix B.4. We can observe that no approach consistently outperforms the other approaches in terms of the number of communication rounds. However, the communication size of ADCOL is always much smaller than the other approaches. ADCOL can save at least 10 times the communication costs to achieve the same accuracy as FedAvg. The speedup can even be up to $34\times$ on Digits. The results demonstrate that ADCOL is much more communication-efficient than the other FL approaches.

## 5.4 SENSITIVITY STUDIES

We study the effect of different factors on ADCOL and the other baselines using the Digit task.

**Scalability and Heterogeneity**   We adopt the same approach as (Li et al., 2021b) to study the effect of number of parties and heterogeneity. We divide each dataset into ten parts randomly and equally and allocate each part into one party. The parties from the same dataset are treated as IID and the parties from different datasets are treated as non-IID. We add two parties from each dataset each time, which results in the number of parties $N \in \{10, 20, 30, 40, 50\}$. Moreover, the degree of heterogeneity decreases as the number of parties increases since the number of IID parties increases. The test accuracies are reported in Figure 2a. We can observe that the accuracy of all approaches can be slightly improved when increasing the number of parties due to the reduced heterogeneity and increased total amount of data. Given a different number of parties, ADCOL consistently outperforms the other baselines. Although the number of classes to distinguish increases for the discriminator when increasing $N$, ADCOL still shows a good and stable performance with a large number of parties.

**Number of Local Epochs**   We vary the number of local epochs $E \in \{1, 2, 5, 10, 20\}$ and report the results in Figure 2b. We run all approaches for 100 rounds. If the number of local epochs is too small, the local update is small in each round and the convergence speed is slow. Thus, the accuracy of all approaches is relatively low after running for 100 rounds with a small number of local epochs. ADCOL still consistently outperforms the other approaches with a different number of epochs.

**Effect of Local Dataset Size**   We vary the percentage of the original local dataset used in each party from 20% to 100%. The results are shown in Figure 2c. The improvement of ADCOL is more

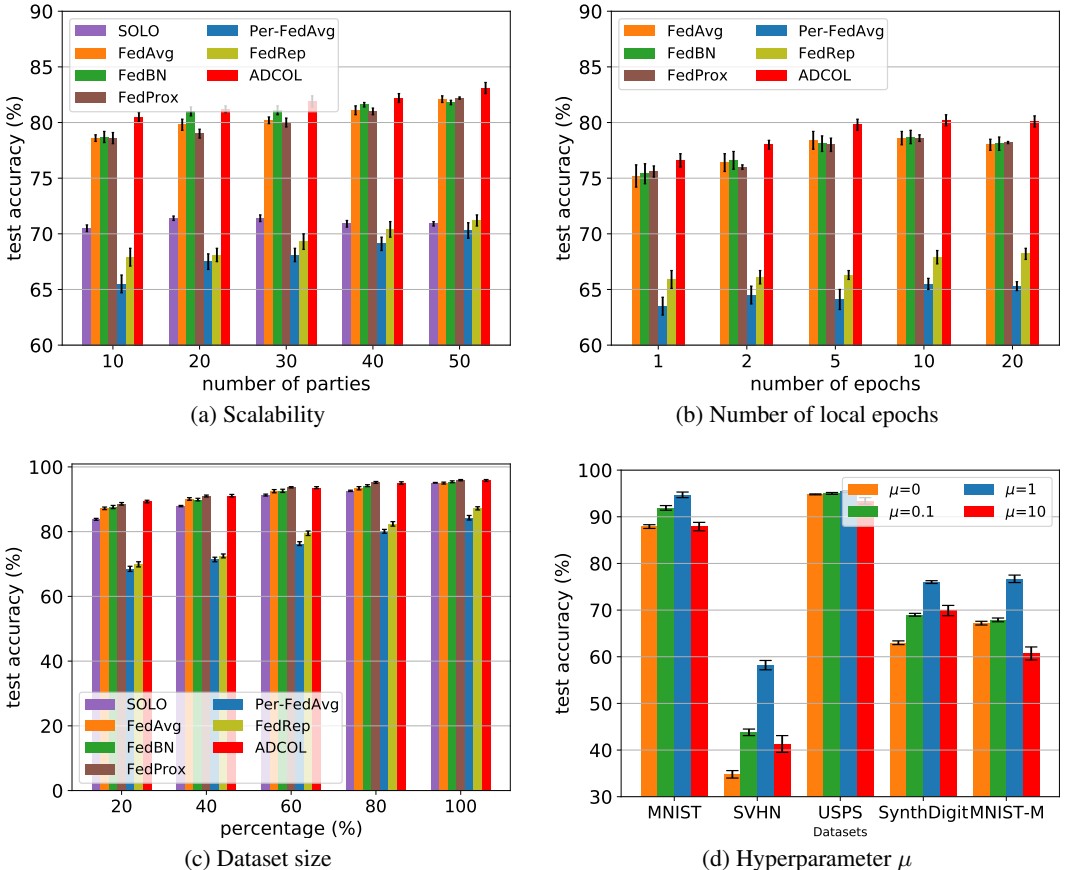

Figure 2: Effect of different factors. We run three trials and report the mean accuracy across parties and its standard derivation.

significant when the size of the local dataset is small. If the size of the local dataset is large, each party can already achieve satisfactory accuracy by SOLO. The accuracy of all approaches is close when the percentage is 100%. It is not necessary to conduct collaborative learning in such a case.

**Effect of** $\mu$  We vary $\mu \in \{0, 0.1, 1, 10\}$ and report the accuracy of ADCOL as shown in Figure 2d. We can observe that ADCOL can achieve the best accuracy when $\mu = 1$. If $\mu$ is too small, the KL divergence loss of Equation (2) has little effect on the local training. Then, the goal of learning a common representation distribution may not achieve. If $\mu$ is too large, the cross-entropy loss of Equation (2) has little effect on the local training, and the representations may not be useful for classification at all (e.g., all representations are a constant vector). ADCOL with $\mu = 10$ may even be worse than SOLO (i.e., $\mu = 0$). Thus, an appropriate $\mu$ is important in ADCOL. Through our experimental studies, we find that setting $\mu = 1$ is a good default choice.

## 6 CONCLUSION

In this paper, we propose ADCOL, a novel collaborative learning approach for non-IID features. ADCOL trains the models in an adversarial way between the parties and the server from a perspective of representation distributions. The parties aim to learn a common representation distribution, while the server aims to distinguish the representations by party IDs. Our experiments on three real-world tasks show that ADCOL achieves higher accuracy than the other state-of-the-art federated learning approaches on non-IID features. ADCOL provides new insights for collaborative learning.

**Ethics Statement**  Not applicable.

**Reproducibility Statement** We have provided the experimental details in Section 5.1 and Section B.1 for reproducibility. Moreover, we will make the code publicly available.

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

## A   THEORETICAL ANALYSIS

**Theorem 1.** We use $P_{G_i}$ to denote the distribution of the representations generated in party $i$ and $P_{G_i}(\mathbf{z})$ is the probability of representation $\mathbf{z}$ in distribution $P_{G_i}$. Then, the optimal discriminator $D^*$ of Equation (4)

$$D_k^*(\mathbf{z}) = \frac{P_{G_k}(\mathbf{z})}{\sum_{i=1}^N P_{G_i}(\mathbf{z})}. \tag{7}$$

*Proof.* From the view of the distribution of representations $\mathbf{z}$, we can reformulate Equation (4) and the objective is to maximize:

$$\sum_{i=1}^N \int_{\mathbf{z}} P_{G_i}(\mathbf{z}) \log(D_i(\mathbf{z})) d\mathbf{z} \tag{8}$$

Let $V(D) = \sum_{i=1}^N P_{G_i}(\mathbf{z}) \log(D_i(\mathbf{z}))$. Note that $\sum_{i=1}^N D_i(\mathbf{z}) = 1$. Let $F(D) = V(D) + \lambda(1 - \sum_{i=1}^N D_i(\mathbf{z}))$. We have

$$\frac{\partial F(D)}{\partial D_i(\mathbf{z})} = \frac{P_{G_i}(\mathbf{z})}{D_i(\mathbf{z})} - \lambda \tag{9}$$

Let $\frac{\partial F(D)}{\partial D_i(\mathbf{z})} = 0$ for $i \in [1, N]$, we have

$$\frac{P_{G_1}(\mathbf{z})}{D_1(\mathbf{z})} = \frac{P_{G_2}(\mathbf{z})}{D_2(\mathbf{z})} = \cdots = \frac{P_{G_N}(\mathbf{z})}{D_N(\mathbf{z})} = \lambda \tag{10}$$

Thus, $V(D)$ can achieve maximum when

$$D_k^*(\mathbf{z}) = \frac{P_{G_k}(\mathbf{z})}{\sum_{i=1}^N P_{G_i}(\mathbf{z})} \tag{11}$$

$\square$

**Theorem 2.** Given the optimal discriminator $D^*$ from Equation (7), the global minimum of Equation (3) is achieved if and only if

$$P_{G_1} = P_{G_2} = \cdots = P_{G_N} \tag{12}$$

*Proof.* From Equation (3), the local objective of party $k$ is to minimize

$$
\begin{aligned}
W(G_k) &= -\mathbb{E}_{\mathbf{x} \sim \mathcal{D}_k} \frac{1}{N} \sum_{i=1}^N \log(N \cdot D_i(G_k(\mathbf{x}))) \\
&= -\mathbb{E}_{\mathbf{x} \sim D_k} \frac{1}{N} \sum_{i=1}^N \log\left(\frac{N \cdot P_{G_i}(G_k(\mathbf{x}))}{\sum_{j=1}^N P_{G_j}(G_k(\mathbf{x}))}\right)
\end{aligned}
\tag{13}
$$

Note that $\sum_{i=1}^N \frac{P_{G_i}(G_k(\mathbf{x}))}{\sum_{j=1}^N P_{G_j}(G_k(\mathbf{x}))} = 1$. Similar to the proof in Theorem 1, Equation (13) can achieve minimum when

$$\frac{P_{G_1}(G_k(\mathbf{x}))}{\sum_{j=1}^N P_{G_j}(G_k(\mathbf{x}))} = \frac{P_{G_2}(G_k(\mathbf{x}))}{\sum_{j=1}^N P_{G_j}(G_k(\mathbf{x}))} = \cdots = \frac{P_{G_N}(G_k(\mathbf{x}))}{\sum_{j=1}^N P_{G_j}(G_k(\mathbf{x}))} = \frac{1}{N}. \tag{14}$$

For $\forall k \in [1, N]$ and $\forall i \in [1, N]$, we have $P_{G_i}(G_k(\mathbf{x})) = \frac{\sum_{j=1}^N P_{G_j}(G_k(\mathbf{x}))}{N}$. Given a representation $\mathbf{z}$, we have

$$P_{G_1}(\mathbf{z}) = P_{G_2}(\mathbf{z}) = \cdots = P_{G_N}(\mathbf{z}) = \frac{\sum_{j=1}^N P_{G_j}(\mathbf{z})}{N} \tag{15}$$

Thus, $P_{G_1} = P_{G_2} = \cdots = P_{G_N}$.

$\square$

Table 3: The statistics of all studied datasets.

|  |  | #training samples | #testing samples |
|---|---|---|---|
| Digits | MNIST | 56,000 | 14,000 |
|  | SVHN | 79,431 | 19,858 |
|  | USPS | 7,438 | 1,860 |
|  | SynthDigit | 402,209 | 97,791 |
|  | MNIST_M | 56,000 | 14,000 |
| Caltech-10 | Amazon | 766 | 192 |
|  | Caltech | 898 | 225 |
|  | DSLR | 125 | 32 |
|  | WebCam | 236 | 59 |
| DomainNet | Clipart | 2,103 | 526 |
|  | Infograph | 2,626 | 657 |
|  | Painting | 2,472 | 619 |
|  | Quickdraw | 4,000 | 1,000 |
|  | Real | 4,864 | 1,217 |
|  | Sketch | 2,213 | 554 |

**Theorem 3.** Suppose $P_G^*$ is the optimal solution shown in Theorem 2. If $G_i$ ($\forall i \in [1, N]$) and $D$ have enough capacity, and $P_{G_i}$ is updated to minimize the local objective (i.e., Equation (3)), given the optimal discriminator $D^*$ from Equation (7), then $P_{G_i}$ converges to $P_G^*$.

*Proof.* In Equation (13), consider $W(G_k) = U(P_{G_i})$ as a function of $P_{G_i}$. Then

$$\frac{\partial U(P_{G_i})}{\partial P_{G_i}} = -\frac{\sum_{k \neq i} P_{G_k}}{N P_{G_i}(P_{G_i} + \sum_{k \neq i} P_{G_k})}. \tag{16}$$

We have

$$\frac{\partial^2 U(P_{G_i})}{\partial P_{G_i}{}^2} = \frac{1}{P_{G_i}^2} - \frac{1}{(P_{G_i} + \sum_{k \neq i} P_{G_k})^2} \geq 0 \tag{17}$$

Thus, $U(P_{G_i})$ is convex in $P_{G_i}$. Therefore, with sufficiently small updates of $P_{G_i}$, $P_{G_i}$ converges to $P_G^*$, concluding the proof. □

## B    ADDITIONAL EXPERIMENTAL RESULTS

### B.1    ADDITIONAL EXPERIMENTAL DETAILS

In each experiment, like FedBN (Li et al., 2021b), to remove the effect of quantity skew, we truncate the size of all datasets to their smallest number with random sampling. For Digits, we resize all images to $28 \times 28 \times 3$ and normalize them with mean 0.5 and standard derivation 0.5 for each channel. For Office-Caltech-10, we resize all images to $64 \times 64 \times 3$ with random horizontal flip and random rotation. For DomainNet, we resize all images to $64 \times 64 \times 3$ with random horizontal flip and random rotation. Like FedBN, we take Digits as the benchmark task for most studies.

The statistics of all the datasets are shown in Table 3. The label distributions are shown in Figure 3. The portion of samples with each class is close to 0.1. The label distribution is balanced among the parties. We train a ResNet-50 on all datasets (i.e., parties) from a task and extract the feature distributions of each dataset. Then, we use t-SNE to visualize the representation as shown in Figure 4. We can observe that the feature distribution of each party is different.

### B.2    CALTECH-10 AND DOMAINNET

Table 4 and 5 show the test accuracy of different approaches on Caltech-10 and DomainNet, respectively. We can observe that ADCOL still outperforms the other approaches in most cases.

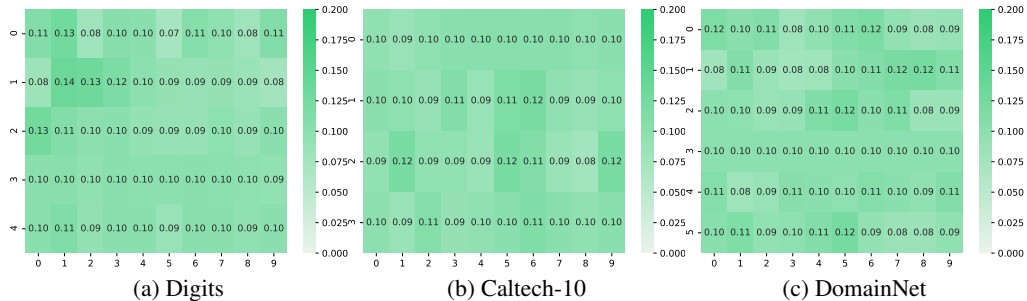

(a) Digits          (b) Caltech-10          (c) DomainNet

Figure 3: The label distributions of each task. The value in each cell of row $i$ and column $j$ represents the percentage of samples with class $j$ in Party $i$.

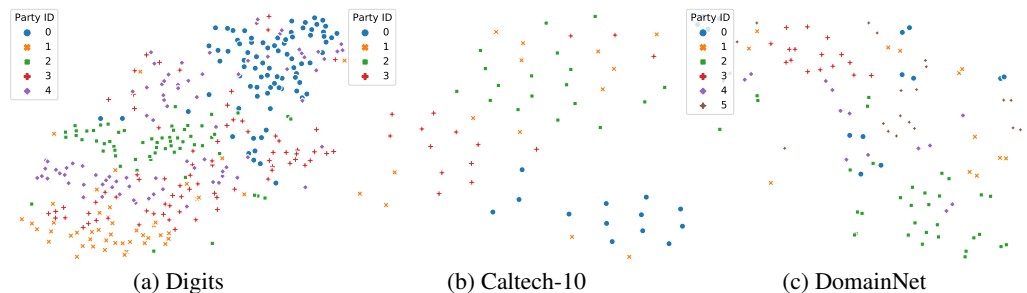

(a) Digits          (b) Caltech-10          (c) DomainNet

Figure 4: The feature distributions of each task.

## B.3 TRAINING CURVES

The training curves of different approaches on Digit are shown in Figure 5. We can observe that ADCOL is much more communication-efficient than the other approaches. ADCOL can convergence with a much smaller communication size than the other approaches.

## B.4 COMMUNICATION EFFICIENCY

Like Table 2, we show the communication efficiency of ADCOL on Caltech-10 and DomainNet in Table 6 and Table 7. We can observe that ADCOL is much more communication-efficient than the other approaches. The speedup can be even up to 300 times.

## B.5 PARTY SAMPLING

Party sampling is a technique usually used in the cross-device setting, where a fraction of parties is sampled to participate in federated learning in each round. Here we set the sample fraction to 0.4 in Digit and choose FedAvg and FedBN as the baselines. The training curves are shown in Figure 6. We can observe that all approaches have an unstable accuracy during training due to sampling. Moreover, FedAvg and FedBN have a very poor accuracy, which shows that existing federated learning approaches cannot well support party sampling on non-IID features. ADCOL significantly outperforms the other approaches.

We increase the number of parties to 100 (i.e., divide each dataset to 20 subsets) and vary the sampling rate from $\{0.1, 0.2, 0.5, 1\}$. We run all approaches for 200 rounds. The results are shown in Table 8. We can observe that when the sampling rate decreases, the performance of all approaches decreases. Moreover, the training is more unstable if the sampling rate is smaller. However, ADCOL still significantly outperforms FedAvg and FedBN. It is still a challenging task to develop effective algorithm on the cross-device setting with a low sampling rate.

Table 4: The comparison of top-1 test accuracy among different approaches on Caltech-10.

| Caltech-10 | Amazon | Caltech | DSLR | WebCam | AVG |
|---|---|---|---|---|---|
| SOLO | 52.8%±0.9% | 36.0%±0.9% | 71.9%±0.5% | 74.6%±0.5% | 58.8%±0.6% |
| FedAvg | 24.0%±1.7% | 36.9%±1.4% | 81.3%±0.4% | 82.7%±0.4% | 56.2%±1.0% |
| FedBN | 33.3%±1.5% | 33.8%±1.8% | 81.3%±0.5% | 80.0%±0.6% | 57.1%±1.0% |
| FedProx | 43.2%±1.6% | 33.1%±0.8% | **82.6%**±0.4% | **83.1%**±0.7% | 60.5%±0.7% |
| Per-FedAvg | 33.9%±1.6% | 32.4%±1.5% | 62.5%±0.8% | 74.6%±0.8% | 50.8%±1.3% |
| FedRep | 16.1%±1.9% | 22.7%±1.7% | 56.3%±1.0% | 57.6%±1.1% | 38.2%±1.4% |
| ADCOL | **54.2%**±1.1% | **38.2%**±1.3% | 75%±0.6% | **83.1%**±0.5% | **62.6%**±0.9% |

Table 5: The comparison of top-1 test accuracy among different approaches on DomainNet.

| DomainNet | Clipart | Infograph | Painting | Quickdraw | Real | Sketch | AVG |
|---|---|---|---|---|---|---|---|
| SOLO | 31.7%±0.9% | 20.2%±1.2% | 30.9%±0.9% | 48.2%±1.1% | 36.5%±0.8% | 20.8%±1.5% | 31.4%±0.9% |
| FedAvg | 33.5%±1.2% | 20.4%±0.7% | 29.2%±0.8% | 56.2%±1.2% | 40.5%±0.8% | 22.6%±1.1% | 33.7%±0.9% |
| FedBN | 36.3%±1.3% | 20.4%±0.8% | 27.8%±1.4% | 61.3%±0.9% | 41.9%±1.3% | 23.6%±1.2% | 35.2%±1.1% |
| FedProx | 37.8%±1.0% | 21.6%±0.8% | 28.1%±1.0% | 23.6%±0.7% | **43.6%**±1.2% | 22.4%±1.0% | 29.5%±1.0% |
| Per-FedAvg | 38.2%±0.7% | 20.2%±0.6% | 27%±1.2% | 42.4%±0.8% | 40.3%±0.7% | 22.6%±1.2% | 31.8%±1.3% |
| FedRep | 27.9%±1.2% | 19%±0.9% | 24.1%±1.2% | 17.9%±1.1% | 31.1%±1.4% | 16.8%±1.1% | 22.8%±1.4% |
| ADCOL | **39.9%**±1.0% | **21.9%**±0.9% | **33.9%**±1.2% | **61.7%**±0.8% | 39.3%±1.5% | **23.9%**±1.2% | **36.8%**±1.0% |

## B.6 STUDY ON THE DISCRIMINATOR

One natural question is how to increase the information contained in the discriminator to improve the performance of ADCOL. We have tried two approaches.

**Changing Model Architecture** One approach is to increase the capacity of the discriminator. We change the model architecture to ResNet-50. The results are shown in Table 9. The performance of ADCOL cannot be improved by increasing the capacity of the discriminator.

**Increasing the Number of Discriminators** The other one approach is to increase the number of discriminators. Suppose the number of discriminators is $N_d$ and the current round is $t$. Then, we use discriminators from round $\max(1, t - N_d)$ to round $(t - 1)$ in the local training. The KL divergence loss is computed as

$$\ell = \frac{1}{N_d} \sum_{i=1}^{N_d} \ell_{KL}([\frac{1}{N}]^N \mid\mid D^i(G(\mathbf{x}))), \tag{18}$$

where $D^i$ is the discriminator trained in round $\max(1, t - i)$. The results are shown in Table 10. ADCOL cannot benefit from more discriminators. When the number of discriminators is larger, the accuracy of ADCOL is even worse. It is a future work to investigate how to integrate more useful information into the discriminator.

## B.7 DIMENSION OF REPRESENTATIONS

Same as SimSam (Chen and He, 2021), we set the dimension of representations (i.e., the output dimension of the projection head, the input dimension of the discriminator) to 2048 by default. As shown in Table 11, we report the performance of ADCOL varying the representation dimension. ADCOL can benefit from a larger representation dimension, where the representations are more informative. The mean accuracy can be improved by about 5% by increasing the dimension from 512 to 2048.

## B.8 NON-IID LABELS

We test the performance of ADCOL on non-IID label settings. Specifically, we sample $p_k \sim Dir_N(0.5)$ and allocate a $p_{k,j}$ proportion of the instances of class $k$ to party $j$, where $Dir(0.5)$ is the Dirichlet distribution with a concentration parameter 0.5. The results are shown in Table 12. ADCOL cannot achieve a better performance than FedAvg. Intuitively, the task-specific representations of images from different classes should be very different. If the label distribution varies across parties, the representation distribution naturally also varies a lot. The intuition of ADCOL, which aims to learn a common representation distribution, is not appropriate on non-IID label settings.

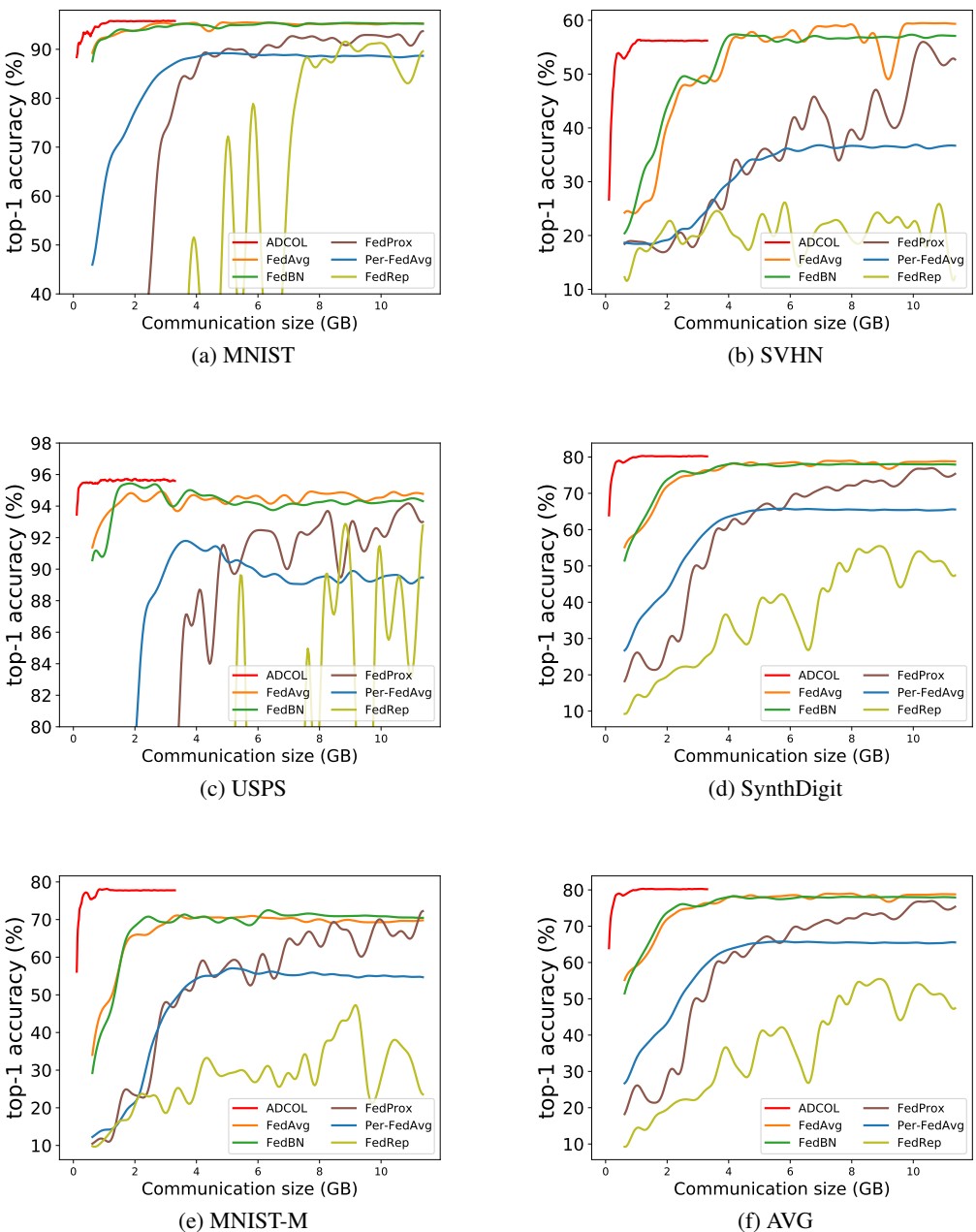

Figure 5: The training curves of different approaches on Digit.

## B.9 IMBALANCED LOCAL DATA SIZES

In the main paper, we truncate the local data to the same size to control non-related factors. Here we show experiments with imbalanced number of samples across parties. We use 10% original data size for all parties without truncating the data. The results are shown in Table 13. ADCOL is still the best approach in most cases.

Table 6: The communication round and communication cost of each approach to achieve the same target performance on Caltech-10.

| Caltech-10 | | Amazon | Caltech | DSLR | WebCam | AVG |
|---|---|---|---|---|---|---|
| #round | FedAvg | 7 | 22 | 12 | 14 | 14 |
| | FedBN | 3 | 21 | 12 | 19 | 19 |
| | FedProx | 14 | 29 | 15 | 29 | 22 |
| | Per-FedAvg | 23 | ╲ | ╲ | ╲ | ╲ |
| | FedRep | ╲ | ╲ | ╲ | ╲ | ╲ |
| | ADCOL | 3 | 18 | 31 | 12 | 20 |
| cost (GB) | FedAvg | 1.99 | 6.25 | 3.41 | 3.98 | 3.98 |
| | FedBN | 0.85 | 5.96 | 3.41 | 5.40 | 5.40 |
| | FedProx | 3.98 | 8.24 | 4.26 | 8.24 | 6.25 |
| | Per-FedAvg | 6.53 | ╲ | ╲ | ╲ | ╲ |
| | FedRep | ╲ | ╲ | ╲ | ╲ | ╲ |
| | ADCOL | 0.02 | 0.10 | 0.17 | 0.07 | 0.11 |
| Speedup | | **120.48** | **63.11** | **19.99** | **60.24** | **36.15** |

Table 7: The communication round and communication cost of each approach to achieve the same target performance on DomainNet.

| DomainNet | | Clipart | Infograph | Painting | Quickdraw | Real | Sketch | AVG |
|---|---|---|---|---|---|---|---|---|
| #round | FedAvg | 22 | 5 | 9 | 38 | 23 | 17 | 47 |
| | FedBN | 14 | 7 | 5 | 54 | 29 | 18 | 41 |
| | FedProx | 13 | 8 | 16 | ╲ | 32 | 12 | ╲ |
| | Per-FedAvg | 31 | 19 | 34 | ╲ | 64 | 57 | ╲ |
| | FedRep | ╲ | ╲ | ╲ | ╲ | ╲ | ╲ | ╲ |
| | ADCOL | 11 | 4 | 9 | 6 | 44 | 19 | 26 |
| cost (GB) | FedAvg | 6.25 | 1.42 | 2.56 | 10.79 | 6.53 | 4.83 | 13.35 |
| | FedBN | 3.98 | 1.99 | 1.42 | 15.34 | 8.24 | 5.11 | 11.64 |
| | FedProx | 3.69 | 2.27 | 4.54 | ╲ | 9.09 | 3.41 | ╲ |
| | Per-FedAvg | 8.80 | 5.40 | 9.66 | ╲ | 18.18 | 16.19 | ╲ |
| | FedRep | ╲ | ╲ | ╲ | ╲ | ╲ | ╲ | ╲ |
| | ADCOL | 0.06 | 0.02 | 0.05 | 0.04 | 0.26 | 0.11 | 0.15 |
| Speedup | | **96.9** | **60.6** | **48.5** | **306.9** | **25.3** | **43.4** | **87.6** |

## B.10 COMPUTATION OVERHEAD

As shown in Table 14, the training time of ADCOL is larger than the other approaches. ADCOL requires the training of a discriminator in the server side, while the other approaches only need to average the models in the server side. However, in practice, the server usually has much powerful computation resources than the parties. Thus, the computation overhead in the server side is affordable.

## B.11 SHARING THE PREDICTOR LAYERS

The parties only send the representations to the server in ADCOL. While ADCOL aims to learn a common representation distribution $z$, an interesting extension is to share the predictor layers between

Table 8: The performance of different approaches varying the sampling rate. We run all approaches for 200 rounds and report the final mean accuracy and standard deviation with three runs.

| Sampling Rate | FedAvg | FedBN | ADCOL |
|---|---|---|---|
| 0.1 | 12.1%±9.2% | 25.3%±8.1% | **34.1%±6.5%** |
| 0.2 | 15.3%±7.6% | 31.9%±7.4% | **46.7%±4.2%** |
| 0.5 | 60.2%±5.4% | 62.2%±4.9% | **67.3%±2.4%** |
| 1 | 77.6%±1.4% | 77.7%±1.2% | **79.1%±0.9%** |

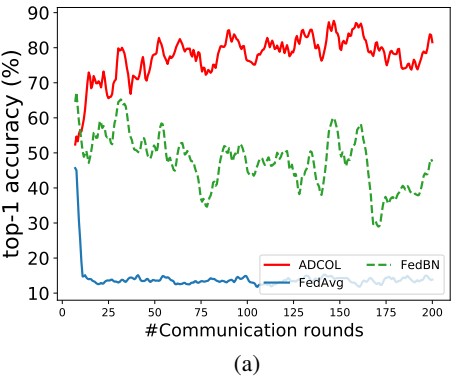 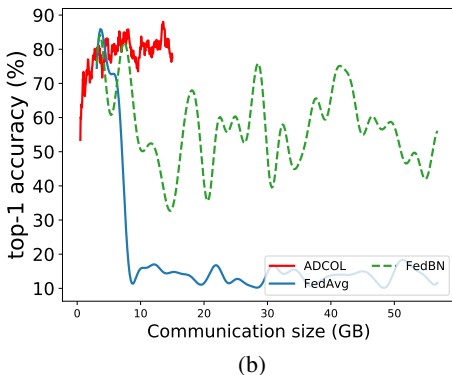

(a)  (b)

Figure 6: The training curves with party sampling (sample fraction = 0.4). We report the mean test accuracy across all parties.

Table 9: ADCOL with different discriminator architectures.

| Discriminator | MNIST | SVHN | USPS | SynthDigit | MNIST-M | AVG |
|---|---|---|---|---|---|---|
| MLP | $95.1\% \pm 0.5\%$ | $55.6\% \pm 0.8\%$ | $96.0\% \pm 0.3\%$ | $73.6\% \pm 0.5\%$ | $76.5\% \pm 0.5\%$ | $79.4\% \pm 0.4\%$ |
| ResNet-50 | $94.7\% \pm 0.6\%$ | $58.2\% \pm 1.0\%$ | $95.4\% \pm 0.2\%$ | $76.0\% \pm 0.3\%$ | $76.7\% \pm 0.8\%$ | $80.2\% \pm 0.5\%$ |

the parties and the server, which ideally helps in regularizing $p(y|\mathbf{z})$. The results are shown in Table 15. We can observe that ADCOL without sharing the predictor layers is generally more effective than sharing the predictor layers. In practice, the distribution $p(y|\mathbf{x}_i)$ is not exactly the same across parties. Thus, it is not necessary to regularize $p(y|\mathbf{z})$ among the parties. Leaving the parties to fine tune their own predictor layer is more capable to learn the personalized local distribution.

### B.12 REPRESENTATION SAMPLING

An approach to reduce the communication cost is to sample a subset of representations and send them to the server. Here we change the sampling rate from $\{20\%, 60\%, 80\%, 100\%\}$. The final accuracy and the communication efficiency are shown in Table 16. We can observe that the communication cost of ADCOL can be significantly reduced with representation sampling technique. Moreover, there is little accuracy loss when the sampling rate is large than $60\%$.

### B.13 DIFFERENTIAL PRIVACY

We conduct experiments to apply differential privacy (Dwork, 2011) to protect the communicated representations. Specifically, we normalize the representations and add noises from $Lap(0, 1/\epsilon)$ before sending them to the server, where $Lap(0, 1/\epsilon)$ is the Laplace distribution with mean 0 and scale $1/\epsilon$. Then, in each round, the transferred representations satisfy $\epsilon-$differential privacy (Lyu et al., 2020). We run 50 rounds for all approaches. For each round, we sample $\frac{1}{50}$ of the representations of each party without replacement so that the privacy cost of each round is not accumulated. The results are shown in Table 17. We can observe that the accuracy of ADCOL is very close to the non-private version with a modest privacy budget (i.e., 10). It is promising to apply differential privacy if there are concerns about the information leakage of the representations.

## C DISCUSSION

**Relation to Domain Adaptation**  Domain adaptation aims to train a model on a source domain (or multi-source domain), which has a good accuracy on a target domain. A classic and popular approach in domain adaptation is to perform adversarial training, i.e., training a discriminator to encourage

Table 10: ADCOL with different number of discriminators.

| Number of discriminators | 1 | 2 | 10 | 20 |
|---|---|---|---|---|
| MNIST | 94.7% ± 0.6% | 95.1% ± 0.6% | 91.0 % ± 0.3% | 88.6% ± 0.8% |
| SVHN | 58.2% ± 1.0% | 45.4% ± 1.1% | 52.4% ± 1.2% | 46.5% ± 1.4% |
| USPS | 95.4% ± 0.2% | 95.2% ± 0.1% | 95.3% ± 0.2% | 90.1% ± 0.4% |
| SynthDigit | 76.0% ± 0.3% | 73.4% ± 0.4% | 67.2% ± 0.8% | 73.4% ± 0.5% |
| MNIST-M | 76.7% ± 0.8% | 76.4% ± 0.7% | 72.1% ± 0.9% | 57.0% ± 1.2% |
| AVG | 80.2% ± 0.5% | 77.1% ± 0.6% | 75.6% ± 0.8% | 71.1% ± 1.0% |

Table 11: The test accuracy of ADCOL with different representation dimensions.

| Dimension | 512 | 1024 | 2048 |
|---|---|---|---|
| MNIST | 93.2% ± 0.6% | 94.9% ± 0.5% | 94.7% ± 0.6% |
| SVHN | 48.9% ± 1.2% | 50.5% ± 1.4% | 58.2% ± 1.0% |
| USPS | 94.8% ± 0.2% | 95.5% ± 0.3% | 95.4% ± 0.2% |
| SynthDigit | 72.9% ± 0.8% | 79.4% ± 0.7% | 76.0% ± 0.3% |
| MNIST-M | 68.9% ± 0.9% | 73.1% ± 0.7% | 76.7% ± 0.8% |
| AVG | 75.7% ± 0.7% | 78.7% ± 0.7% | 80.2% ± 0.5% |

domain-invariant features (Ganin et al., 2016; Peng et al., 2019b). For the difference between our approach and domain adaptation, Peng et al. (2019b) proposed FADA, which extends domain adaptation in a federated setting. One connection between our approach and domain adaptation is that each party can be viewed as a source domain, and the target domain is the unknown oracle optimal. Then, our approach is to extract domain-invariant features from multiple source domain, which is used to regularize the training. To highlight the differences between our approach and the domain adaptation techniques, we compare our approach with the federated domain adaptation study (Peng et al., 2019b) (FADA) and show the main differences: (1) Setting: FADA aims to train a model on multiple source domain, which has a good accuracy on a target domain. Our study aims to train a personalized model for each party, which has good accuracy on its local data. (2) Discriminator: FADA uses multiple discriminators, where each discriminator is used for binary classification for one source-target domain pair. Our study uses a single discriminator for the multi-classification among all parties. Moreover, we have provided the theoretical analysis on the convergence properties. (3) Framework: FADA uses adversarial training to generate domain-invariant and domain-specific features. Our study uses adversarial training to regularize the local training in federated learning.

Intuitively, we cannot directly compare ADCOL and FADA in the experiments since the settings are different. In our experiments, there is no a single target domain for testing in FADA. One method is to treat each party as a target domain and applying FADA $N$ times, where $N$ is the number of parties. However, the computation and communication overhead is significantly large. Moreover, such an approach does not utilize the labels of the target dataset. We have compared ADCOL and FADA using the above method and the results are shown in Table 18. ADCOL significantly outperforms FADA. Moreover, the test accuracy of FADA is even smaller than local training in many cases since it does not exploit the labels of the target dataset.

**Limitations** ADCOL is a collaborative learning method for non-IID features. As shown in Appendix B.8, the performance of ADCOL is poor compared with federated learning approaches on non-IID label setting. Note that ADCOL aims to learn a common representation distribution. Intuitively, the task-specific representations of images from different classes should be very different, which can be easily classified by a small MLP. Thus, if the label distribution varies across parties, the representation distribution naturally also varies a lot across parties. The current objective of ADCOL does not fit into the non-IID label setting.

As shown in Section 3.5, the communication size of ADCOL is related to the number of examples. If the number of examples is very large and the size of the model is small, the communication cost of ADCOL will be larger than other federated learning approaches. However, local training can usually achieve satisfactory performance if the dataset size is very large. In such cases, besides ADCOL, existing federated learning approaches may also not help.

Table 12: The test accuracy of different approaches on non-IID label settings.

|  | SOLO | FedAvg | ADCOL |
|---|---|---|---|
| CIFAR-10 | $59.3\% \pm 8\%$ | $89.0\% \pm 2.4\%$ | $79.2\% \pm 4.7\%$ |
| CIFAR-100 | $33.5\% \pm 2.1\%$ | $57\% \pm 2.8\%$ | $36.1\% \pm 2.8\%$ |

Table 13: The performance of differerent approaches in the imbalanced data size setting.

|  | SOLO | FedAvg | FedBN | ADCOL | FedProx | Per-FedAvg | FedRep |
|---|---|---|---|---|---|---|---|
| MNIST | 93.9% | 98.1% | 98.0% | 98.7% | 96.7% | 94.5% | 97.6% |
| SVHN | 84.2% | 88.3% | 88.4% | 87.1% | 69.1% | 76.7% | 86.5% |
| USPS | 94.2% | 93.1% | 93.6% | 95.6% | 94.0% | 92.6% | 94.1% |
| SynthDigit | 94.0% | 95.4% | 95.6% | 96.1% | 93.7% | 90.0% | 95.6% |
| MNIST_M | 90.9% | 89.7% | 92.4% | 94.2% | 86.0% | 82.5% | 89.4% |
| AVG | 91.4% | 92.9% | 93.6% | 94.3% | 87.9% | 87.3% | 92.6% |

Table 14: The total training time of running all approaches for 100 rounds.

|  | FedAvg | FedBN | ADCOL | FedProx | Per-FedAvg | FedRep |
|---|---|---|---|---|---|---|
| Total Training time (hour) | 6.5 | 7 | 11 | 8 | 7 | 8.5 |

Table 15: The comparison between sharing the predictor layers and not sharing the predictor layers.

|  | w/ sharing predictor layers | w/o sharing predictor layers |
|---|---|---|
| MNIST | **95.4**% | 94.7% |
| SVHN | 48.8% | **58.2**% |
| USPS | 95.2% | **95.4**% |
| SynthDigit | 75.9% | **76.0**% |
| MNIST_M | 73.4% | **76.7**% |
| AVG | 77.7% | **80.2**% |

Table 16: ADCOL with different representation sampling rates. We present the final converged mean accuracy and the number of communication rounds and communication costs to achieve the target accuracy 78%.

| Sampling rate | accuracy | #communication rounds | communication cost (GB) |
|---|---|---|---|
| 20% | 78.4% | 32 | 0.07 |
| 60% | 79.1% | 26 | 0.17 |
| 80% | 79.8% | 23 | 0.20 |
| 100% | 80.2% | 21 | 0.23 |

Table 17: Differentially private ADCOL.

| privacy budget | non-private | 2 | 5 | 10 |
|---|---|---|---|---|
| MNIST | 94.7% | 89.1% | 92.7% | 93.1% |
| SVHN | 58.2% | 50.4% | 56.7% | 56.9% |
| USPS | 95.4% | 88.2% | 93.0% | 93.1% |
| SynthDigit | 76.0% | 64.1% | 69.4% | 73.2% |
| MNIST_M | 76.7% | 66.8% | 71.9% | 74.3% |
| AVG | 80.2% | 71.7% | 76.7% | 78.1% |

**Insights and Future Work**    The key insights from ADCOL are (1) a GAN-style training scheme and (2) regularization from a view of representation distribution. While ADCOL does not have a requirement on the vanilla local training algorithm, it can also be extended to self-supervised federated learning, where the cross-entropy loss is replaced by the loss used in self-supervised learning (e.g., contrastive loss (Chen et al., 2020; Chen and He, 2021)). Moreover, while ADCOL only works on non-IID feature settings currently, the adversarial collaborative training scheme can potentially be

Table 18: The comparison between SOLO, ADCOL, and FADA on Digits.

|       | MNIST | SVHN  | USPS  | SynthDigit | MNIST_M | AVG   |
|-------|-------|-------|-------|------------|---------|-------|
| SOLO  | 87.9% | 34.8% | 94.8% | 63.0%      | 67.2%   | 69.5% |
| ADCOL | 94.7% | 58.2% | 95.4% | 76.0%      | 76.7%   | 80.2% |
| FADA  | 85.6% | 40.1% | 89.6% | 68.8%      | 60.5%   | 68.9% |

applied to address other data settings by modifying the objectives of local training and server training. There are many research opportunities based on the findings of this paper.

