# OpenReview forum: "Adversarial Collaborative Learning on Non-IID Features"
_ICLR.cc/2022/Conference — ICLR 2022 Submitted_

### Official Review · Reviewer_VD2k · 2021-11-01

**Correctness:** 3
**Technical Novelty And Significance:** 2
**Empirical Novelty And Significance:** 2
**Recommendation:** 5
**Confidence:** 4

**Main Review:**

Strengths:
1). This paper studies an important problem -- the non-iid features problem in federated learning. The authors have some good ideas to tackle the problem.
2). The paper presents both theoretical and numerical results. The experiments show that the proposed method has some advantages over some baselines.
3). The paper is well-written in general.

Weaknesses:
1).  Using adversarial learning to add the non-iid features problem in federated learning is not new. The proposed method is similar to the FedUFO algorithm in  "Federated Learning for Non-IID Data via Unified Feature Learning and Optimization Objective Alignment" [Zhang et al. ICCV'21] (not cited). The authors might want to compare the proposed method with FedUFO.

2). The proposed method requires the server to distinguish the devices' local representation.  In practice, only a small number of devices are selected for each round of federated learning. In addition, the number of the selected devices might also be different. It is questionable whether the devices can learn a common representation efficiently under these scenarios. Even if they can, these dynamics might significantly degrade the performance of the ADCOL algorithm. The authors might want to explain why the proposed method can be applied to these scenarios. The appendix has shown some results related to device sampling, but it only has one large sampling rate (0.4). In practice, the sampling rate is much lower (e.g., below 0.2).

3). While the experiments appear to be comprehensive, there is a lack of study on how different degrees of non-iid influence the proposed algorithm's performance. Besides, the authors might want to increase the number of devices and adjust the device sampling rate.


**Summary Of The Paper:**

This paper studies federated learning when different devices have non-iid features. To address the heterogeneity problem,  it proposes a federated learning scheme called ADCOL based on adversarial learning.  In ADCOL, the devices transfer local representations to the server while sending the discriminator to the devices. The server aims to distinguish the devices' local representations, while the devices aim to train local models that generate non-distinguishable representations. To make the representations non-distinguishable, ADCOL adds an additional regularization term to the devices' loss functions. This practice aims to maximize the probability that the discriminator cannot distinguish the local representations. The experimental results show that the proposed method has some advantages over several baselines.

**Summary Of The Review:**

While this paper has some nice ideas to address the non-iid feature problems in federated learning, its contributions are limited due to novelty and practicability issues.

---

> ### Author Response · Authors · 2021-11-22
> **Response to Reviewer VD2k**
>
> Thanks for all your comments. We have addressed them as below.
>
> * Using adversarial learning to add the non-iid features problem in federated learning is not new. The proposed method
> is similar to the FedUFO algorithm in "Federated Learning for Non-IID Data via Unified Feature Learning and Optimization
> Objective Alignment" [Zhang et al. ICCV'21] (not cited). The authors might want to compare the proposed method with FedUFO.
>
>
> > We have added the reference in Section 2.2 of the revised paper. We highlight that FedUFO [1] is a contemporary work.
> The accepted papers of ICCV'21 are only available in Oct 2021, which is after the ICLR 2022 submission deadline.
> Note that FedUFO is based on FedAvg, which still applies a model-averaging training scheme. Our approach's training framework is based on adversarial training,
> which is significantly different from FedUFO. Moreover, the communication overhead of FedUFO is large since each party needs to send its
> local model to all the other parties. The experiments of FedUFO are based on non-IID label distribution, while our approach focuses on non-IID feature distribution.
> Also, the code of FedUFO is not open-sourced. We have asked the authors for code and there is no reply yet.  Thus, we are not able to compare our approach with FedUFO now.
>
> > [1] Federated Learning for Non-IID Data via Unified Feature Learning and Optimization Objective Alignment.  ICCV'21.
>
>
>
> * The proposed method requires the server to distinguish the devices' local representation. In practice, only a small number of devices are selected for each round of federated learning. In addition, the number of the selected devices might also be different. It is questionable whether the devices can learn a common representation efficiently under these scenarios. Even if they can, these dynamics might significantly degrade the performance of the ADCOL algorithm. The authors might want to explain why the proposed method can be applied to these scenarios. The appendix has shown some results related to device sampling, but it only has one large sampling rate (0.4). In practice, the sampling rate is much lower (e.g., below 0.2).
>
>
> > We agree that the performance of ADCOL will be affected if the sampling rate is very low. In the earlier rounds,
> the discriminator may not be well updated if the classes of the received representations are limited.
> However, the performance of FedAvg also decreases significantly. Under the non-IID feature distribution, if only selecting
> a small number of parties for training, then it is like training the model in a very biased subset.
> We have increased the number of parties to 100 and varied the sampling rate from $\{0.1, 0.2, 0.5\}$.
> The results are shown in Table 8 of the revised paper. We can observe that when the sampling rate decreases,
> the performance of all approaches decreases. Moreover, the training is more unstable if the sampling rate is smaller.
> However, ADCOL still significantly outperforms FedAvg and FedBN. It is still a challenging task to develop an effective algorithm on the cross-device setting with a low sampling rate.
>
> * While the experiments appear to be comprehensive, there is a lack of study on how different degrees of non-iid influence the proposed algorithm's performance. Besides, the authors might want to increase the number of devices and adjust the device sampling rate.
>
> > While it is challenging to directly modify the feature distribution of the datasets to control the degree of non-IID,
> we use the same approach as in [2] to control the degree of non-IID. As shown in the second paragraph of Section 5.4, we divide each dataset into ten parts randomly and equally and allocate each part into one party.
> The parties from the same dataset are treated as IID and the parties from different datasets are treated as non-IID. We add two parties from each dataset each time.
> Then, the degree of heterogeneity decreases as the number of parties increases since the number of IID parties increases.
> As shown in Figure 2 (a), the performance of ADCOL increases as the degree of non-IID decreases. As stated in the above response, for the experiments on increasing
> the number of parties and adjusting the sampling rate, please refer to Table 8.
>
> > [2] FEDBN: FEDERATED LEARNING ON NON-IID FEATURES VIA LOCAL BATCH NORMALIZATION. ICLR 2021.

---

### Official Review · Reviewer_tbKF · 2021-11-02

**Correctness:** 3
**Technical Novelty And Significance:** 2
**Empirical Novelty And Significance:** 2
**Recommendation:** 3
**Confidence:** 5

**Main Review:**

Strengths:
(1) It proposed a novel adversarial collaborative learning for federated learning on non-IID features.
(2) It provided the convergence state of the training process.
(3) Experiments showed it outperformed baselines.

Weaknesses:
(1) The overall idea is marginally novel, as it simply adapts the training procedures of “domain-adversarial neural network” [ref 1] into the federated learning. This idea has also been explored in previous work [ref 2]. In this case, [ref 2] also focused on the non-IID features in FL. It is more convincing to use [ref 2] as one of the baselines in the experiments.
[ref 1] Ganin, Yaroslav, Evgeniya Ustinova, Hana Ajakan, Pascal Germain, Hugo Larochelle, François Laviolette, Mario Marchand, and Victor Lempitsky. "Domain-adversarial training of neural networks." The journal of machine learning research 17, no. 1 (2016): 2096-2030.
[ref 2] Peng, Xingchao, Zijun Huang, Yizhe Zhu, and Kate Saenko. "Federated Adversarial Domain Adaptation." In International Conference on Learning Representations. 2020.
(2) One major concern of this method is to the privacy issue. It suffers much more seriously from the privacy leakage by sharing the hidden representations, compared to sharing only parameters in FedAvg. This might break the privacy constraint of federated learning. The communication cost also significantly increases due to the shared representations.
(3) The theoretical results provided the convergence state of the training process, but it is not clear whether the proposed FL method can achieve the convergence result.
(4) The train/test split for each client is not given.


**Summary Of The Paper:**

This work presented an adversarial collaborative learning named ADCOL for federated learning on non-IID features. It learns the invariant feature representations for all parties.

**Summary Of The Review:**

The novelty of the proposed model is incremental, and the privacy issue of sharing features is not solved.

---

> ### Author Response · Authors · 2021-11-22
> **Response to Reviewer tbKF**
>
> Thanks for all your comments. We have addressed them as below.
>
> * The overall idea is marginally novel, as it simply adapts the training procedures of “domain-adversarial training of neural network” [ref 1] into the federated learning. This idea has also been explored in previous work [ref 2]. In this case, [ref 2] also focused on the non-IID features in FL. It is more convincing to use [ref 2] as one of the baselines in the experiments. [ref 1] Ganin, Yaroslav, Evgeniya Ustinova, Hana Ajakan, Pascal Germain, Hugo Larochelle, François Laviolette, Mario Marchand, and Victor Lempitsky. "Domain-adversarial training of neural networks." The journal of machine learning research 17, no. 1 (2016): 2096-2030. [ref 2] Peng, Xingchao, Zijun Huang, Yizhe Zhu, and Kate Saenko. "Federated Adversarial Domain Adaptation." In International Conference on Learning Representations. 2020.
>
>
> > We highlight that our approach is different from adversarial domain adaptation in the aspects of setting, discriminator, and framework. We have added the clarification in the first paragraph of Appendix C (Page 19).
> >> (1) Setting: FADA [2] aims to train a model on multiple source domains, which has a good accuracy on a target domain. Our study aims to train a personalized model for each party,
> which has good accuracy on its local data. In our setting, there is no a single target domain for training and testing.
> Thus, we **cannot directly compare ADCOL and FADA** in the experiments since the settings are different.
> One method is to treat each party as a target domain and applying FADA N times, where N is the number of parties.
> However, the computation and communication overhead is significantly large.
> Moreover, such an approach does not utilize the labels of the target dataset.
> We have compared ADCOL and FADA using the above method and the results are shown in Table 18 of the revised paper. ADCOL significantly outperforms FADA.
> Moreover, **the test accuracy of FADA is even smaller than local training in many cases since it does not exploit the labels of the target dataset**.
>
> >> [1] Ganin, Yaroslav, Evgeniya Ustinova, Hana Ajakan, Pascal Germain, Hugo Larochelle, François Laviolette, Mario Marchand, and Victor Lempitsky. "Domain-adversarial training of neural networks." The journal of machine learning research 17, no. 1 (2016): 2096-2030.
>
> >> [2] Peng, Xingchao, Zijun Huang, Yizhe Zhu, and Kate Saenko. "Federated Adversarial Domain Adaptation." In International Conference on Learning Representations. 2020.
>
> >> (2) Discriminator: [2] uses multiple discriminators, where each discriminator is used for binary classification for one source-target domain pair.
> Our study uses a single discriminator for the multi-classification among all parties. Moreover, we have provided the theoretical analysis of the convergence properties.
>
> >> (3) Framework: [2] uses adversarial training to generate domain-invariant and domain-specific features.
> Our study uses adversarial training to regularize the local training in federated learning.
>
>
> * One major concern of this method is to the privacy issue. It suffers much more seriously from the privacy leakage by sharing the hidden representations, compared to sharing only parameters in FedAvg. This might break the privacy constraint of federated learning.
> The communication cost also significantly increases due to the shared representations.
>
> > Please refer to 'Response to the Common Concerns'.
>
> * The theoretical results provided the convergence state of the training process, but it is not clear whether the proposed FL method can achieve the convergence result.
>
> > Like other studies on GANs [3,4], the theoretical results aim to give insights on the convergence
> of our approach, which aims to demonstrate that the optimal representation distribution of each party is same ideally
> given the proposed KL divergence loss term. In practice, we optimize the parameter $\theta$ of the local networks rather than the distribution $P_{G_i}$ itself.
> Thus, it is challenging to show the gap between the theoretical results and the empirical results.
>
>
> > [3] Generative Adversarial Nets. NeurIPS 2014.
>
> > [4] Self-supervised GAN: Analysis and Improvement with Multi-class Minimax Game. NeurIPS 2019.
>
> * The train/test split for each client is not given.
>
> > For each dataset, we randomly split 1/5 of the original dataset as the test dataset, while the remained dataset is used as the training dataset. We have added it in the Setup paragraph of Page 7.

---

> > ### Comment · Reviewer_tbKF · 2021-11-28
> > **Thank you for the response!**
> >
> > I would like to keep my score unchanged for the following reasons. First, the techniques proposed in this work are very similar to FADA and original DANN. In FADA or DANN, one can simply let the target domain be the mixed sources. In this case, they would minimize the distribution discrepancy of any pair of sources. The major difference of ADCOL and FADA is that ADCOL uses the multi-classification as the discriminator. But the multi-classification discriminator has also used in DANN when there are multiple sources in some work [1-2]. Second, it is not convincing for me to share the features in the FL setting. Compared to the raw data, the learned features are much more representative. To some extent, it is easier to uncover the properties (e.g., sampling distribution, class membership) from data features compared to learning from the raw data. Third, the convergence of FL on Non-IID Data has been recently studied in [3]. A similar theoretical convergence analysis of the overall ADCOL training will be more convincing.
> >
> > [1] Matsuura, Toshihiko, and Tatsuya Harada. "Domain generalization using a mixture of multiple latent domains." In Proceedings of the AAAI Conference on Artificial Intelligence, vol. 34, no. 07, pp. 11749-11756. 2020.
> > [2] Sicilia, Anthony, Xingchen Zhao, and Seong Jae Hwang. "Domain adversarial neural networks for domain generalization: When it works and how to improve." arXiv preprint arXiv:2102.03924 (2021).
> > [3] Li, Xiang, Kaixuan Huang, Wenhao Yang, Shusen Wang, and Zhihua Zhang. "On the Convergence of FedAvg on Non-IID Data." In International Conference on Learning Representations. 2020.

---

> > > ### Author Response · Authors · 2021-11-29
> > > **Further Response to Reviewer tbKF**
> > >
> > > Thanks for your comments. We think you may overlook that the raw data are not allowed to exchange in the federated setting.
> > > * In FADA or DANN, one can simply let the target domain be the mixed sources.
> > >
> > > > We cannot let the target domain be the mixed sources. Note that the target generator in FADA needs to be updated by accessing the target data (see Eq. 4 and Eq. 5 of the FADA paper). In the federated setting, **we cannot mix the source data since the raw data cannot be transferred**.
> > >
> > > * The major difference of ADCOL and FADA is that ADCOL uses the multi-classification as the discriminator.
> > >
> > > > As we have mentioned, the settings and the frameworks of FADA and ADCOL are totally different. FADA and ADCOL aim to solve different problems. As we have shown in Table 18, FADA has a poor performance in our setting. If you only focus on distribution matching by adversarial learning, the technique has been used in many studies (e.g., GANs [1, 2]) besides domain adaptation. We do not think these studies are very similar.
> > >
> > > > [1] Generative Adversarial Networks. NeurIPS 2014
> > >
> > > > [2] Adversarially Learned Inferences. ICLR 2017
> > >
> > > * Compared to the raw data, the learned features are much more representative. To some extent, it is easier to uncover the properties (e.g., sampling distribution, class membership) from data features compared to learning from the raw data.
> > >
> > > > We are not sure what do you mean by learning from the raw data. Apparently, the raw data leak more information than the representations. The sampling distribution and class membership are directly available from the raw data. Moreover, sharing raw data is not allowed in the federated setting.
> > >
> > > > If you refer to exchanging models like FedAvg. There have been many studies [3,4,5] on attacks on the models, which can infer the class-wised representatives and even the raw data. Differential privacy is a de facto approach to provide privacy guarantees. We have shown the applicability of differential privacy in ADCOL in Appendix B.13. Although we do not mainly focus on the privacy issue in this paper, we appreciate it if you can provide any references that compare the vulnerability of representations and models so that we can add them in our future revision.
> > >
> > > > [3] Model Inversion Attacks that Exploit Confidence Information and Basic Countermeasures
> > >
> > > > [4] Deep Models Under the GAN: Information Leakage from Collaborative Deep Learning
> > >
> > > > [5] Comprehensive privacy analysis of deep learning: Stand-alone and federated learning under passive and active white-box inference attacks
> > >
> > > * The convergence of FL on Non-IID Data has been recently studied in [6]. A similar theoretical convergence analysis of the overall ADCOL training will be more convincing.
> > >
> > > > As we have claimed, ADCOL is conducted in a GAN-like framework, which is not based on distributed SGD. Thus, we analyze the convergence properties like the studies in GANs [7,8]. We appreciate it if you can provide any GAN-style studies that provide the convergence analysis like [6].
> > >
> > > > [6] On the Convergence of FedAvg on Non-IID Data
> > >
> > > > [7] Generative Adversarial Nets. NeurIPS 2014.
> > >
> > > > [8] Self-supervised GAN: Analysis and Improvement with Multi-class Minimax Game. NeurIPS 2019.

---

### Official Review · Reviewer_gd59 · 2021-11-02

**Correctness:** 4
**Technical Novelty And Significance:** 3
**Empirical Novelty And Significance:** 3
**Recommendation:** 8
**Confidence:** 4

**Main Review:**

Pros:

- The paper addresses an important problem, namely the one of federated learning that accounts for the fact the real-world data for collaborative learning is often non-iid in the features.
- The paper is well-written and motivated, comparison to prior work is very well-done.
- The proposed adversarial training method is intuitive and seems to perform well in practice.
- The issues of communication complexity and privacy are discussed, even if those can in some situations be limitations of the proposed method.

Cons:

- It is indeed possible that for certain modes (for example the local sample sizes being large as compared to the number of users N), the proposed method can be communication-inefficient.

- Similarly, privacy can be seen as an issue. It is true that classic FL is also prone to privacy problems, however the proposed method sends multiple features, of the same input data, to the server at every round. Intuitively, it should be easier to infer information about the original data with so much information, rather than just with sharing (one) locally updated model at every round.

Suggestions:

- The paper will benefit from further explanation on why the proposed method (finding representation that obfuscates the differences of the distributions of the local datasets) is expected to work. The idea is intuitive and is in lines with classic domain adaptation techniques. However, expanding the discussion on that will be beneficial.

- Similarly, equation (1) and the KL distance come a bit out of nowhere in the text. It will be nice to justify using the KL, as opposed to other distance measures.

- In Section 3.1, it will be useful to repeat again the assumption that only the input distribution changes, since this is where you formally introduce the studied problem.

- The conclusion (and in some sense the comparison to standard FL throughout the paper) seems to suggest that the method is more computationally efficient than other FL methods. The validity of this claim clearly depends on the values of multiple parameters of the problem, such as n, N, d etc. In this sense, the technical discussion is Section 3.5 is much more helpful to the reader and I would recommend against overselling the merits of the proposed method.

- Related to the last point, perhaps the authors could comment on the possibility of communicating the representations of only a subset of the local data of each client at every round?

**Summary Of The Paper:**

The paper proposes an adversarial training method for learning from non-iid data in federated learning. Both a theoretical and an experimental analysis of the method are provided.

**Summary Of The Review:**

Overall, this is a very well-written paper that studies an important topic and proposes an interesting new idea. While there are some limitations, they are quite well-discussed (although improvements such as those mentioned in the "Suggestions" section are desirable) and I think that the paper will spark some interesting discussions and new ideas in the context of FL.

---

> ### Author Response · Authors · 2021-11-22
> **Response to Reviewer gd59**
>
> Thanks for all your comments. We have addressed them as below.
>
> * It is indeed possible that for certain modes (for example the local sample sizes being large as compared to the number of users N), the proposed method can be communication-inefficient.
>
> > Please refer to 'Response to the Common Concerns'.
>
> * Privacy can be seen as an issue.
>
> > Please refer to 'Response to the Common Concerns'.
>
>
>
> * The paper will benefit from further explanation on why the proposed method (finding representation that obfuscates
> the differences of the distributions of the local datasets) is expected to work. The idea is intuitive and is in lines with classic domain adaptation techniques. However, expanding the discussion on that will be beneficial.
>
> > We have added the discussion on the comparison between our method and domain adaptation in the first paragraph of Appendix C (Page 19).
> A classic and popular approach in domain adaptation is to perform adversarial training, i.e., training a discriminator to encourage domain-invariant features.
> In our approach, we train a discriminator to encourage the learning of the task-specific sufficient representation distribution among parties.
> One connection between our approach and domain adaptation is that each party can be viewed as a source domain, and the target domain
> is the unknown oracle optimal. Then, our approach is to extract domain-invariant features from multiple source domains, which is used to regularize the training. For the difference between our approach and domain adaptation,
> please refer to the first paragraph of Appendix C.
>
>
>
> * Similarly, equation (1) and the KL distance come a bit out of nowhere in the text. It will be nice to justify using the KL, as opposed to other distance measures.
>
> > Our method aims to measure the distance between different probability distributions. Thus, KL divergence is the most suitable one, which has statistical meaning (i.e., measure of information loss between distributions) while other distances such as L2 distance do not have. We have added the clarification in the second paragraph of Section 3.2.
>
> * In Section 3.1, it will be useful to repeat again the assumption that only the input distribution changes, since this is where you formally introduce the studied problem.
>
> > We have added the assumption that the label distribution is similar among parties in Section 3.1.
>
> * The conclusion (and in some sense the comparison to standard FL throughout the paper) seems to suggest that the method is more computationally efficient than other FL methods. The validity of this claim clearly depends on the values of multiple parameters of the problem, such as n, N, d etc. In this sense, the technical discussion is Section 3.5 is much more helpful to the reader and I would recommend against overselling the merits of the proposed method.
>
> > We have removed the claim on the communication efficiency of our approach in the conclusion.
>
> * Related to the last point, perhaps the authors could comment on the possibility of communicating the representations of only a subset of the local data of each client at every round?
>
> > Thanks for the suggestion. We have added the experiments to show the proposed approach in Appendix B.12 of the revised paper. Indeed, by applying the representation
> sampling, the communication costs can be reduced while the final accuracy is similar to the vanilla version.

---

### Official Review · Reviewer_3uWW · 2021-11-02

**Correctness:** 3
**Technical Novelty And Significance:** 3
**Empirical Novelty And Significance:** 3
**Recommendation:** 5
**Confidence:** 4

**Main Review:**

### strengths:
1. The motivation of ADCOL (Adversarial Collaborative Learning) is both clear and reasonable. Indeed, the features in real-world scenarios are usually non-IID and how to apply federated learning on the non-IID data has always been an important question in practice. Therefore, it is quite necessary and challenging to design a thorough federated framework.
2. The experimental evaluation seems to be convincing. Authors choose several typical fundamental tasks (digits, office-caltech-10, DomainNet) for experimental evaluation. The results demonstrated the effectiveness of the proposed ADCOL method.

### weaknesses:
1. In ADCOL, parties deliver representations of data instead of local models. It may lead to more serious privacy breaches. It needs more work to protect the local data.
2. The advantage of communication size is not persuasive. In fact, smaller models than used in SOLO are often used
3. The statements in this paper are not complete, especially in the experiment sections. There is a lot of confusion here, such as the statics of all the datasets,  the method for averaging results from different data sources (arithmetic average or weighted average according to data volume)  and methods for adjusting consistent label distribution but non-IID feature distribution, and the final distribution of label and feature.


**Summary Of The Paper:**

This paper focuses on federated learning on non-IID features. This is a crucial problem when applying federated learning. The authors propose a new federated learning scheme, called ADCOL (Adversarial Collaborative Learning) for non-IID features. Specifically, the server is designed to train a discriminator to distinguish the local representations from local parties. While the local parties train the local models and expect the representations not to be distinguished by the discriminator. Authors conduct experiments on multiple parties where the data have heterogeneous features but share the same labels and label distribution and the results clearly show the effectiveness.

**Summary Of The Review:**

As shown in the Main Review, the motivation of the paper is clear and reasonable, and also, the experimental evaluations are convincing. While there are also some issues that need further refinement to be accepted, especially the statements in the experiment.

---

> ### Author Response · Authors · 2021-11-22
> **Response to Reviewer 3uWW**
>
> Thanks for all your comments. We have addressed them as below.
>
> * In ADCOL, parties deliver representations of data instead of local models. It may lead to more serious privacy breaches. It needs more work to protect the local data.
>
> > Please refer to 'Response to the Common Concerns'.
>
> * The advantage of communication size is not persuasive. In fact, smaller models than used in SOLO are often used.
>
> > Please refer to 'Response to the Common Concerns'.
>
>
> * The statements in this paper are not complete, especially in the experiment sections. There is a lot of confusion here, such as the statics of all the datasets, the method for averaging results from different data sources (arithmetic average or weighted average according to data volume)
> and methods for adjusting consistent label distribution but non-IID feature distribution, and the final distribution of label and feature.
>
> > We have added more details in the experiments.
> >> Statics: The statics of all the datasets are available in Table 3 of the revised paper.
>
> >> Method for averaging: Like FedAvg, we use weighted average according to data volume for all studied approaches.
> We have added it in the first paragraph of Section 5.1.
>
> >> Methods for adjusting consistent label distribution but non-IID feature distribution: As stated in Setup of Page 7,
> for each task, each party has data from one of the data sources. Same as [1], the feature distributions are naturally
> different and label distributions are same among parties. We have drawn the label and feature distributions as shown in
> Figure 3 and Figure 4 in Appendix B.1. We can observe that the label distributions are balanced among the parties and the feature distributions are different among the parties.
>
>
> >> [1] FedBN: Federated Learning on Non-IID Features via Local Batch Normalization    ICLR'21

---

### Author Response · Authors · 2021-11-22
**Response to the Common Concerns**

There are two common concerns of the reviewers: privacy and communication overhead of sharing representations.

## Privacy

A common concern from the reviewers is the privacy of sending representations in our approach. We address the privacy concern from three aspects.
> **We have conducted experiments to apply differential privacy to protect the communicated representations as shown in Appendix B.13 of the revised paper**. Specifically, we normalize the representations and add noises from $Lap(0, 1/\epsilon)$,
before sending them to the server, where $Lap(0, 1/\epsilon)$ is the Laplace distribution with mean 0 and scale $1/\epsilon$. Then, in each round, the transferred representations satisfy $\epsilon-$differential privacy [1].
We run 50 rounds for all approaches. For each round, we sample $\frac{1}{50}$ of the representations of each party without replacement so that the privacy cost of each round is not accumulated. The results are shown in Table 17 of the revised paper.
We can observe that the accuracy of ADCOL is very close to the non-private version with a modest privacy budget (i.e., 10). It is promising to apply differential privacy if there are concerns about the information leakage of the representations.

> [1] Differentially private representation for nlp: Formal guarantee and an empirical study on privacy and fairness. EMNLP 2020.

>To the best of our knowledge, there is no existing study that theoretically or empirically compares the vulnerability between representations and models.
As we have stated in the Privacy paragraph of Page 5, the comparison between the privacy leakage between representations and models is a
future direction, which is out of the scope of this paper. There is no proof that the representation contains more information, and vice verse.
Actually, existing studies have shown that **the mutual information between the input data and the final representation is very small** [2].

> [2] Opening the Black Box of Deep Neural Networks via Information.

> **Sharing representations has been used in existing federated learning studies** [3,4]. According to the study [5], federated learning only requires
that the raw data is not exchanged and there is no restriction on the format of the exchanged information.

> [3] Group Knowledge Transfer: Federated Learning of Large CNNs at the Edge. NeurIPS 2020.

> [4] Federated Adversarial Domain Adaptation. ICLR 2020.

> [5] Advances and open problems in federated learning. Foundations and Trends in Machine Learning 2021.

## Communication overhead

Another common concern from the reviewers is that the communication size of ADCOL may be large compared with FedAvg in some cases potentially (e.g., smaller models).
We address the communication issue from three aspects.

> Usually, a smaller model also implies a smaller number of samples as the model capacity is limited.
Let us consider the experimental setting in FedAvg [5] as another example. A simple MLP (199,210 parameters) is used for classification on MNIST with 100 parties.

> [5] Communication-Efficient Learning of Deep Networks from Decentralized Data. AISTATS, 2017.

>>For FedAvg, the communication overhead per round is (2\*100\*4\*199210) B = 159.4 MB.

>>For ADCOL, considering the dimension of representation d=128, and the discriminator as a 2-layer MLP with 128 hidden units ($S_d$ = 68,628 B). Then, the communication
overhead per round of ADCOL is (4\*60000\*128+100\*68628) B = 37.6 MB.

>**The communication cost of ADCOL is still much smaller than FedAvg in a smaller model setting**.

> From the Communication Size paragraph in Page 5, we can observe that ADCOL has more communication than FedAvg
when the number of parties is smaller, the number of samples is larger, and the size of model is smaller. However, in such a setting,
**it is impractical to conduct federated learning as local training may already achieve satisfactory performance**.
Moreover, when the number of parties is small, it is very likely the cross-silo setting, where the parties are organizations and the communication
is less important than the cross-device setting (e.g., mobile devices).
While there are potentially some cases that our approach is less communication-efficient than FedAvg, we show that
ADCOL is communication-efficient in the common experimental settings.

> **As suggested by Reviewer gd59, we have added a technique to further reduce the communication overhead**.
We sample the representations and send the sampled subset to the server in each round. The results are shown in Appendix B.12.
The communication cost can be further reduced by representation sampling.

---

### Author Response · Authors · 2021-11-29
**Summary of Changes**

Here is the summary of the main changes in the revision.
> We have added the comparison between ADCOL with FedUFO in Section 2.2.

> We have explained why adopting KL divergence in Section 3.2.

> We have added the study on the effect of heterogeneity in Section 5.4.

> We have added more details about the experiments, including dataset statistics, the method for averaging results, and the visualization results of the feature and label distributions (Appendix B.1).

> We have added the party sampling experiment with a larger number of parties and a low sampling rate in Appendix B.5.

> We have added the representation sampling technique in Appendix B.12.

> We have added the experiments to apply differential privacy when sharing the representations in Appendix B.13.

> We have added the comparison between ADCOL with domain adaptation in Appendix C.

---

### Decision · Program_Chairs · 2022-01-20

**Decision:**

Reject

**Comment:**

This manuscript proposes an adversarial method to address non-IID heterogeneity on federated learning. Distinct from existing methods, the mitigation is implemented by training and local communicating learned representations, so the metric of success is indistinguishability of representations across devices.

There are four reviewers, all of whom agree that the method addresses an interesting and timely issue. However, reviewers are mixed on the paper score, and many raised concerns about communication overhead, apparent privacy costs, and convergence concerns with the adversarial methods. There is also some limited concern of novelty compared to existing methods. The authors provide a good rebuttal addressing these issues -- either based on experimental evidence (adding differential privacy), comparing communication overhead tradeoffs as it varies with model and sample size, and some existing convergence analysis. However, after reviews and discussion, the reviewers are unconvinced that the method is sufficiently robust to the concerns outlined. Authors are encouraged to address the highlighted technical concerns in any future submission of this work.